# DIFFERENTIALLY PRIVATE BILEVEL OPTIMIZATION

## ABSTRACT

We present differentially private (DP) algorithms for bilevel optimization, a problem class that received significant attention lately in various machine learning applications. These are the first DP algorithms for this task that are able to provide any desired privacy, while also avoiding Hessian computations which are prohibitive in large-scale settings. Under the well-studied setting in which the upper-level is not necessarily convex and the lower-level problem is strongly-convex, our proposed gradient-based $(\epsilon, \delta)$-DP algorithm returns a point with hypergradient norm at most $\widetilde{\mathcal{O}}\left((\sqrt{d_{\mathrm{up}}}/\epsilon n)^{1/2} + (\sqrt{d_{\mathrm{low}}}/\epsilon n)^{1/3}\right)$ where $n$ is the dataset size, and $d_{\mathrm{up}}/d_{\mathrm{low}}$ are the upper/lower level dimensions. Our analysis covers constrained and unconstrained problems alike, accounts for mini-batch gradients, and applies to both empirical and population losses.

## 1 INTRODUCTION

Bilevel optimization is a fundamental framework for solving optimization objectives of hierarchical structure, in which constraints are defined themselves by an auxiliary optimization problem. Formally, it is defined as

$$\text{minimize}_{\mathbf{x} \in \mathcal{X}} \quad F(\mathbf{x}) := f(\mathbf{x}, \mathbf{y}^*(\mathbf{x})) \tag{BO}$$
$$\text{subject to} \quad \mathbf{y}^*(\mathbf{x}) \in \arg\min_{\mathbf{y}} g(\mathbf{x}, \mathbf{y}) \,,$$

where $F : \mathbb{R}^{d_x} \to \mathbb{R}$ is referred to as the hyperobjective, $f : \mathbb{R}^{d_x} \times \mathbb{R}^{d_y} \to \mathbb{R}$ as the upper-level (or outer) objective, and $g : \mathbb{R}^{d_x} \times \mathbb{R}^{d_y} \to \mathbb{R}$ as the lower-level (or inner) objective. While bilevel optimization is well studied for over half a century (Bracken & McGill, 1973), it has recently received significant attention due to its diverse applications in machine learning (ML). These include hyperparameter tuning (Bengio, 2000; Maclaurin et al., 2015; Franceschi et al., 2017; 2018; Lorraine et al., 2020), meta-learning (Andrychowicz et al., 2016; Bertinetto et al., 2018; Rajeswaran et al., 2019; Ji et al., 2020), neural architecture search (Liu et al., 2018), invariant learning (Arjovsky et al., 2019; Jiang & Veitch, 2022), and data reweighting (Grangier et al., 2023; Fan et al., 2024; Pan et al., 2024). In these applications, both the upper and lower level objectives in (BO) typically represent some loss over data, and are given by empirical risk minimization (ERM) problems with respect to some dataset $S = \{\xi_1, \ldots, \xi_n\} \in \Xi^n$ :[1]

$$f(\mathbf{x}, \mathbf{y}) := f_S(\mathbf{x}, \mathbf{y}) = \frac{1}{n} \sum_{i=1}^{n} f(\mathbf{x}, \mathbf{y}, \xi_i) \,, \qquad g(\mathbf{x}, \mathbf{y}) := g_S(\mathbf{x}, \mathbf{y}) = \frac{1}{n} \sum_{i=1}^{n} g(\mathbf{x}, \mathbf{y}, \xi_i) \,, \tag{ERM}$$

often serving as empirical proxies of the stochastic (population) objectives with respect to a distribution $\mathcal{P}$ supported on $\Xi$ :

$$f(\mathbf{x}, \mathbf{y}) := f_{\mathcal{P}}(\mathbf{x}, \mathbf{y}) = \mathbb{E}_{\xi \sim \mathcal{P}}[f(\mathbf{x}, \mathbf{y}; \xi)] \,, \qquad g(\mathbf{x}, \mathbf{y}) := g_{\mathcal{P}}(\mathbf{x}, \mathbf{y}) = \mathbb{E}_{\xi \sim \mathcal{P}}[g(\mathbf{x}, \mathbf{y}; \xi)] \,. \tag{Pop}$$

In this work, we study bilevel optimization under differential privacy (DP) (Dwork et al., 2006). As ML models are deployed in an ever-growing number of applications, protecting the privacy of the

---

[1]It is possible for the datasets with respect to $f$ and $g$ to be distinct (e.g., validation and training data), perhaps of different sizes. We will assume without loss of generality that $S$ is the entire dataset, and thus $n$ is the total number of samples. Concretely, letting $f(\cdot; \xi_i) = 0$ or $g(\cdot; \xi_i) = 0$ for certain indices in order to exclude corresponding data points from either objective, will not affect our results.

data on which they are trained is a major concern, and DP has become the gold-standard for privacy preserving ML (Abadi et al., 2016). Accordingly, DP optimization is extensively studied, with a vast literature focusing both on empirical and stochastic objectives under various assumptions (Chaudhuri et al., 2011; Kifer et al., 2012; Bassily et al., 2014; Wang et al., 2017; Bassily et al., 2019; Wang et al., 2019; Feldman et al., 2020; Tran & Cutkosky, 2022; Gopi et al., 2022; Arora et al., 2023; Carmon et al., 2023; Ganesh et al., 2024; Lowy et al., 2024).

Nonetheless, to the best of our knowledge, no first-order algorithm (i.e., which uses only gradient queries) that solves bilevel optimization problems under DP, is known to date. This is no coincidence: until recently, no first-order methods with finite time guarantees were known even for non-private bilevel problems. This follows the fact (Ghadimi & Wang, 2018, Lemma 2.1) that under mild regularity assumptions, the so-called hypergradient takes the form:

$$\nabla F(\mathbf{x}) = \nabla_x f(\mathbf{x}, \mathbf{y}^*(\mathbf{x})) - \nabla^2_{xy} g(\mathbf{x}, \mathbf{y}^*(\mathbf{x}))[\nabla^2_{yy} g(\mathbf{x}, \mathbf{y}^*(\mathbf{x}))]^{-1} \nabla_y f(\mathbf{x}, \mathbf{y}^*(\mathbf{x})) \ . \tag{1}$$

Consequently, directly applying a "gradient" method to $F$ requires inverting Hessians of the lower level problem at each time step, thus limiting applicability in contemporary high-dimensional applications. Following various approaches to tackle this challenge (see Section 1.2), recent breakthroughs were finally able to provide fully first-order methods for bilevel optimization with non-asymptotic guarantees (Liu et al., 2022; Kwon et al., 2023; Yang et al., 2023; Chen et al., 2024). These recent algorithmic advancements show promising empirical results in large scale applications, even up to the LLM scale of $\sim 10^9$ parameters (Pan et al., 2024). We therefore make use of these techniques for the sake of private optimization. As we will see, our privacy analysis requires overcoming some subtle challenges due to privacy leaking between the inner and outer problems, since $\nabla F(\mathbf{x})$ itself depends on $\mathbf{y}^*(\mathbf{x})$ (seen in Eq. (1)).

The only prior method we are aware of for DP bilevel optimization was recently proposed by Chen & Wang (2024), which falls short in two main aspects. First, their algorithm only provides *some* privacy guarantee which cannot be controlled by the user. Moreover, it requires inverting local Hessians at each step, which significantly limits scalability; see Section 1.2 for further discussion.

## 1.1 OUR CONTRIBUTIONS

We present DP algorithms that solve bilevel optimization problems whenever the the upper level is smooth but not necessarily convex, and the lower level problem is smooth and strongly-convex. To the best of our knowledge, these are the first algorithms to do so using only first-order (i.e. gradient) queries of the upper- and lower-level objectives, and that can ensure any desired privacy $\epsilon, \delta > 0$. Our contributions can be summarized as follows:

- **Bilevel ERM Algorithm (Theorem 3.1):** We present a $(\epsilon, \delta)$-DP first-order algorithm for the bilevel ERM problem (BO/ERM) that outputs with high probability a point with hypergradient norm bounded by

$$\|\nabla F_S(\mathbf{x})\| = \widetilde{\mathcal{O}}\left( \left( \frac{\sqrt{d_x}}{\epsilon n} \right)^{1/2} + \left( \frac{\sqrt{d_y}}{\epsilon n} \right)^{1/3} \right) \ .$$

Our algorithm also adapts to the case where $\mathcal{X} \subsetneq \mathbb{R}^{d_x}$ is a non-trivial constraint set, which is common in certain applications.[2] In the constrained setting, we obtain the same guarantee as above in terms of the projected hypergradient (see Section 2 for details).

- **Mini-batch Bilevel ERM Algorithm (Theorem 4.1):** Aiming for a more practical algorithm, we design a variant of our previous algorithm that relies on mini-batch gradients. For the bilevel ERM problem (BO/ERM), given any batch sizes $b_{\mathrm{in}}, b_{\mathrm{out}} \in \{1, \ldots, n\}$ for sampling gradients of the inner/outer problems respectively, our algorithm ensures $(\epsilon, \delta)$-DP and outputs with high probability a point with hypergradient norm bounded by

$$\|\nabla F_S(\mathbf{x})\| = \widetilde{\mathcal{O}}\left( \left( \frac{\sqrt{d_x}}{\epsilon n} \right)^{1/2} + \left( \frac{\sqrt{d_y}}{\epsilon n} \right)^{1/3} + \frac{1}{b_{\mathrm{out}}} \right) \ . \tag{2}$$

---

[2]For instance, in data reweighting $\mathcal{X}$ is the probability simplex; in hyperparameter tuning it is the hyperparameter space, which is typically constrained.

Notably, Eq. (2) is independent of the inner-batch size, yet depends on the outer-batch size, which coincides with known results for "single"-level constrained nonconvex optimization (Ghadimi et al., 2016) (see Remark 4.2 for further discussion). Our mini-batch algorithm is also applicable in the constrained setting $\mathcal{X} \subsetneq \mathbb{R}^{d_x}$ with the same guarantee in terms of projected hypergradient.

- **Population loss guarantees (Theorem 5.1):** We further provide guarantees for stochastic objectives. In particular, we show that for the population bilevel problem (BO/Pop), our $(\epsilon, \delta)$-DP algorithm outputs with high probability a point with hypergradient norm bounded by

$$\|\nabla F_{\mathcal{P}}(\mathbf{x})\| = \widetilde{\mathcal{O}}\left( \left(\frac{\sqrt{d_x}}{\epsilon n}\right)^{1/2} + \left(\frac{d_x}{n}\right)^{1/2} + \left(\frac{\sqrt{d_y}}{\epsilon n}\right)^{1/3} \right),$$

with an additional additive $1/b_{\text{out}}$ factor in the mini-batch setting.

## 1.2 RELATED WORK

Bilevel optimization was introduced by Bracken & McGill (1973), and grew into a vast body of work, with classical results focusing on asymptotic guarantees for specific problem structures (Anandalingam & White, 1990; Ishizuka & Aiyoshi, 1992; White & Anandalingam, 1993; Vicente et al., 1994; Zhu, 1995; Ye & Zhu, 1997). There exist multiple surveys and books covering various approaches for these problems (Vicente & Calamai, 1994; Dempe, 2002; Colson et al., 2007; Bard, 2013; Sinha et al., 2017).

Ghadimi & Wang (2018) observed Eq. (1) under strong-convexity of the inner problem using the implicit function theorem, therefore asserting that the hypergradient can be computed via inverse Hessians, which requires solving a linear system at each point. Many follow up works built upon this second-order approach with additional techniques such as variance reduction, momentum, Hessian sketches, projection-free updates, or incorporating external constraints (Amini & Yousefian, 2019; Yang et al., 2021; Khanduri et al., 2021; Guo et al., 2021; Ji et al., 2021; Chen et al., 2021; Akhtar et al., 2022; Chen et al., 2022; Tsaknakis et al., 2022; Hong et al., 2023; Jiang et al., 2023; Abolfazli et al., 2023; Merchav & Sabach, 2023; Xu & Zhu, 2023; Cao et al., 2024; Dagréou et al., 2024).

Only recently, the groundbreaking result of Liu et al. (2022) proved finite-time convergence guarantees for a fully first-order method which is based on a penalty approach. This result was soon extended to stochastic objectives (Kwon et al., 2023), with the convergence rate later improved by (Yang et al., 2023; Chen et al., 2024), and also extended to constrained bilevel problems (Yao et al., 2024; Kornowski et al., 2024). The first-order penalty paradigm also shows promise for some bilevel problems in which the inner problem is not strongly-convex (Shen & Chen, 2023; Kwon et al., 2024; Lu & Mei, 2024), which is generally a highly challenging setting (Chen et al., 2024; Bolte et al., 2024). Moreover, Pan et al. (2024) provided an efficient implementation of this paradigm, showing its effectiveness for large scale applications.

As to DP optimization, there is an extensive literature on optimization problems, both for ERM and for stochastic losses, which are either convex (Chaudhuri et al., 2011; Kifer et al., 2012; Bassily et al., 2014; Wang et al., 2017; Bassily et al., 2019; Feldman et al., 2020; Gopi et al., 2022; Carmon et al., 2023) or smooth and nonconvex (Wang et al., 2019; Tran & Cutkosky, 2022; Arora et al., 2023; Ganesh et al., 2024; Lowy et al., 2024).

To the best of our knowledge, the only existing result for DP bilevel optimization is the very recent result of Chen & Wang (2024), which differs than ours in several aspect. Their proposed algorithm is second-order, requiring evaluating Hessians, and solving corresponding linear systems at each time step, which we avoid altogether. Moreover, Chen & Wang (2024) study the *local* DP model (Kasiviswanathan et al., 2011), in which each user (i.e. $\xi_i$) does not reveal its individual information. Due to this more challenging setting, they can only derive guarantees for *some finite* privacy budget $\epsilon < \infty$, even as the dataset size grows $n \to \infty$. We study the common central DP model, in which a trusted curator acts on the collected data and releases a private solution, and thus are able to provide any desired privacy and accuracy guarantees with sufficiently many samples. Our work is the first to study bilevel optimization in this well-studied DP setting. We also note that Fioretto et al. (2021) studied the related problem of DP in Stackelberg games, which are certain bilevel programs which arise in game theory, aiming at designing coordination mechanisms that maintain the individual agents' privacy.

## 2 PRELIMINARIES

**Notation and terminology.** We let bold-face letters (e.g. $\mathbf{x}$) denote vectors, and denote by $\mathbf{0}$ the zero vector (whenever the dimension is clear from context) and by $I_d \in \mathbb{R}^{d \times d}$ the identity matrix. $[n] := \{1, 2, \ldots, n\}$. $\langle \cdot, \cdot \rangle$ denotes the standard Euclidean dot product, and $\|\cdot\|$ denotes either its induced norm for vectors or operator norm for matrices, and $\|f\|_\infty = \sup_{\mathbf{x} \in \mathcal{X}} |f(\mathbf{x})|$ denotes the sup-norm. We denote by $\mathrm{Proj}_{\mathbb{B}(\mathbf{z}, R)}$ the projection onto the closed ball around $\mathbf{z}$ of radius $R$. $\mathcal{N}(\boldsymbol{\mu}, \Sigma)$ denotes a normal (i.e., Gaussian) random variable with mean $\boldsymbol{\mu}$ and covariance $\Sigma$. We use standard big-O notation, with $\mathcal{O}(\cdot)$ hiding absolute constants (independent of problem parameters), $\widetilde{\mathcal{O}}(\cdot)$ also hiding poly-logarithmic terms. We denote $f \lesssim g$ if $f = \mathcal{O}(g)$, and $f \asymp g$ if $f \lesssim g$ and $g \lesssim f$. A function $f : \mathcal{X} \subseteq \mathbb{R}^{d_1} \to \mathbb{R}^{d_2}$ is $L_0$-Lipschitz if for all $\mathbf{x}, \mathbf{y} \in \mathcal{X} : \|f(\mathbf{x}) - f(\mathbf{y})\| \leq L_0 \|\mathbf{x} - \mathbf{y}\|$; $L_1$-smooth if $\nabla f$ exists and is $L_1$-Lipschitz; and $L_2$ Hessian-smooth if $\nabla^2 f$ exists and is $L_2$-Lipschitz (with respect to the operator norm). A twice-differentiable function $f$ is $\mu$-strongly-convex if $\nabla^2 f \succeq \mu I$, denoting by "$\succeq$" the standard PSD ("Loewner") order on matrices.

**Differential privacy.** Two datasets $S, S' \in \Xi^n$ are said to be neighboring, denoted by $S \sim S'$, if they differ by only one data point. A randomized algorithm $\mathcal{A} : \Xi^n \to \mathcal{R}$ is called $(\epsilon, \delta)$ differentially private (or $(\epsilon, \delta)$-DP) for $\epsilon, \delta > 0$ if for any two neighboring datasets $S \sim S'$ and measurable $E \subseteq \mathcal{R}$ in the algorithm's range, it holds that $\Pr[\mathcal{A}(S) \in E] \leq e^\epsilon \Pr[\mathcal{A}(S') \in E] + \delta$ (Dwork et al., 2006). The basic composition property of DP states that the (possibly adaptive) composition of $(\epsilon_0, \delta_0)$-DP- and $(\epsilon_1, \delta_1)$-DP mechanisms, is $(\epsilon_0 + \epsilon_1, \delta_0 + \delta_1)$-DP. We next recall some well known DP basics: advanced composition, the Gaussian mechanism, and privacy amplification by subsampling.

**Lemma 2.1** (Advanced composition, Dwork et al., 2010). *For $\epsilon_0 < 1$, a $T$-fold (possibly adaptive) composition of $(\epsilon, \delta)$-DP mechanisms is $(\epsilon, \delta)$-DP for $\epsilon = \sqrt{2T \log(1/\delta_0)}\epsilon_0 + 2T\epsilon_0^2$, $\delta = (T + 1)\delta_0$.*

**Lemma 2.2** (Gaussian mechanism). *Given a function $h : \Xi^b \to \mathbb{R}^d$, the Gaussian mechanism $\mathcal{M}(h) : \Xi^b \to \mathbb{R}^d$ defined as $\mathcal{M}(h)(S) := h(S) + \mathcal{N}(\mathbf{0}, \sigma^2 I_d)$ is $(\epsilon, \delta)$-DP for $\epsilon, \delta \in (0, 1)$, as long as $\sigma^2 \geq \frac{2 \log(5/4\delta)(\mathcal{S}_h)^2}{\epsilon^2}$, where $\mathcal{S}_h := \sup_{S \sim S'} \|h(S) - h(S')\|$ is the $L_2$-sensitivity of $h$.*

**Lemma 2.3** (Privacy amplification, Balle et al., 2018). *Suppose $\mathcal{M} : \Xi^b \to \mathcal{R}$ is $(\epsilon_0, \delta_0)$-DP. Then given $n \geq b$, the mechanism $\mathcal{M}' : \Xi^n \to \mathcal{R}$, $\mathcal{M}'(S) := \mathcal{M}(B)$ where $B \sim \mathrm{Unif}(\Xi)^b$, is $(\epsilon, \delta)$-DP for $\epsilon = \log(1 + (1 - (1 - 1/n)^b)(e^{\epsilon_0} - 1))$, $\delta = \delta_0$.*

We remark that advanced composition will be used when $\epsilon_0 \lesssim \sqrt{\log(1/\delta_0)/T}$, thus the accumulated privacy scales as $\epsilon \asymp \sqrt{T}\epsilon_0$. Similarly, privacy amplification will be used when $\epsilon_0 \leq 1$, under which the privacy after subsampling scales as $\epsilon \asymp \frac{b\epsilon_0}{n}$ (since $e^{\epsilon_0} - 1 \asymp \epsilon_0$, $(1 - 1/n)^b \asymp \frac{b}{n}$ and $\log(1 + \frac{b}{n}\epsilon_0) \asymp \frac{b}{n}\epsilon_0$).

**Gradient mapping.** Given a point $\mathbf{x} \in \mathbb{R}^d$, and some $\mathbf{v} \in \mathbb{R}^d$, $\eta > 0$, we denote

$$\mathcal{G}_{\mathbf{v}, \eta}(\mathbf{x}) := \frac{1}{\eta} \left( \mathbf{x} - \mathcal{P}_{\mathbf{v}, \eta}(\mathbf{x}) \right) , \quad \mathcal{P}_{\mathbf{v}, \eta}(\mathbf{x}) := \arg\min_{\mathbf{u} \in \mathcal{X}} \left[ \langle \mathbf{v}, \mathbf{u} \rangle + \frac{1}{2\eta} \|\mathbf{u} - \mathbf{x}\|^2 \right] .$$

In particular, given an $L$-smooth function $F : \mathbb{R}^d \to \mathbb{R}$ and $\eta \leq \frac{1}{2L}$, we denote the projected gradient (also known as reduced gradient) and the gradient (or prox) mapping, respectively, as

$$\mathcal{G}_{F, \eta}(\mathbf{x}) := \frac{1}{\eta} \left( \mathbf{x} - \mathcal{P}_{\nabla F, \eta}(\mathbf{x}) \right) , \quad \mathcal{P}_{\nabla F, \eta}(\mathbf{x}) := \arg\min_{\mathbf{u} \in \mathcal{X}} \left[ \langle \nabla F(\mathbf{x}), \mathbf{u} \rangle + \frac{1}{2\eta} \|\mathbf{u} - \mathbf{x}\|^2 \right] .$$

The projected gradient $\mathcal{G}_{F, \eta}(\mathbf{x})$ generalizes the gradient to the possibly constrained setting: for points $\mathbf{x} \in \mathcal{X}$ sufficiently far from the boundary of $\mathcal{X}$, $\mathcal{G}_{F, \eta}(\mathbf{x}) = \nabla F(\mathbf{x})$, namely it simply reduces to the gradient. See the textbooks (Nesterov, 2013; Lan, 2020) for additional details. We will recall a useful fact, which asserts that the mapping $\mathcal{G}_{\mathbf{v}, \eta}(\mathbf{x})$ is non-expansive with respect to $\mathbf{v}$ :

**Lemma 2.4.** *For any $\mathbf{x}, \mathbf{v}, \mathbf{w} \in \mathbb{R}^d$, $\eta > 0 : \|\mathcal{G}_{\mathbf{v}, \eta}(\mathbf{x}) - \mathcal{G}_{\mathbf{w}, \eta}(\mathbf{x})\| \leq \|\mathbf{v} - \mathbf{u}\|$.*

Lemma 2.4 is important in our analysis, since as we will argue later (in Section 3.1), gradient estimates must be *inexact* in the bilevel setting to satisfy privacy, and Lemma 2.4 will allows us to control the error due to this inexactness. Although this result is known (cf. Ghadimi et al. 2016), we reprove it in Appendix C for completeness.

## 2.1 SETTING

We impose the following assumptions, all of which are standard in the bilevel optimization literature.

**Assumption 2.5.** *For (BO) with either (ERM) or (Pop), we assume the following hold:*

   *i. $\mathcal{X} \subseteq \mathbb{R}^{d_x}$ is a closed convex set.*

   *ii. $F(\mathbf{x}_0) - \inf_{\mathbf{x} \in \mathcal{X}} F(\mathbf{x}) \leq \Delta_F$ for some initial point $\mathbf{x}_0 \in \mathcal{X}$.*

   *iii. $f$ is twice differentiable, and $L_1^f$-smooth.*

   *iv. For all $\xi \in \Xi : f(\cdot, \cdot; \xi)$ is $L_0^f$-Lipschitz (hence, so is $f$).*

   *v. $g$ is $L_2^g$-Hessian-smooth, and for all $\mathbf{x} \in \mathcal{X} : g(\mathbf{x}, \cdot)$ is $\mu_g$-strongly-convex.*

   *vi. For all $\xi \in \Xi : g(\cdot, \cdot; \xi)$ is $L_1^g$-smooth (hence, so is $g$).*

As mentioned, these assumptions are standard in the study of bilevel optimization problems and are shared by nearly all of the previous works we discussed. In particular, the strong convexity of $g(\mathbf{x}, \cdot)$ ensures that $\mathbf{y}^*(\mathbf{x})$ is uniquely defined, which is generally required in establishing the regularity of the hyporobjective. Indeed, it is known that dropping this assumption, can, in general, lead to pathological behaviors not amenable for algorithmic guarantees (cf. Chen et al. 2024; Bolte et al. 2024 and discussions therein). For the purpose of differential privacy though, the strong convexity of $g(\mathbf{x}, \cdot)$ raises a subtle issue. As the standard assumption in the DP optimization literature is that the component functions are Lipschitz, which allows privatization of gradients using sensitivity arguments, strongly-convex objectives cannot be Lipschitz over the entire Euclidean space.[3] Therefore, strongly-convex objectives are regularly analyzed in the DP setting under the additional assumption that the domain of interest is bounded. For bilevel problems, the domain of interest for $\mathbf{y}$ is the lower level solution set, thus we impose the following assumption.

**Assumption 2.6.** *There exists a compact set $\mathcal{Y} \subset \mathbb{R}^{d_y}$ with $\{\mathbf{y}^*(\mathbf{x}) : \mathbf{x} \in \mathcal{X}\} \subseteq \mathcal{Y}$, such that for all $\mathbf{x} \in \mathcal{X}, \xi \in \Xi : g(\mathbf{x}, \cdot; \xi)$ is $L_0^g$-Lipschitz over $\mathcal{Y}$.*

**Remark 2.7.** *Note that $\mathrm{diam}(\mathcal{Y}) \leq 2L_0^g/\mu_g$. Indeed, fixing some $\mathbf{x} \in \mathcal{X}$, since $g(\mathbf{x}, \cdot; \xi)$ is $L_0^g$-Lipschitz over $\mathcal{Y}$ for all $\xi \in \Xi$, then so is $g(\mathbf{x}, \cdot)$. Moreover, by $\mu_g$-strong-convexity, we get that for all $\mathbf{y} \in \mathcal{Y} : \mu_g \|\mathbf{y} - \mathbf{y}^*(\mathbf{x})\| \leq \|\nabla_y g(\mathbf{x}, \mathbf{y})\| \leq L_0^g$. Hence $\mathcal{Y} \subseteq \mathbb{B}(\mathbf{y}^*(\mathbf{x}), L_0^g/\mu_g)$, which is of diameter $2L_0^g/\mu_g$.*

Following Assumptions 2.5 and 2.6, we denote $\ell := \max\{L_0^f, L_1^f, L_0^g, L_1^g, L_2^g\}$, $\kappa := \ell/\mu_g$.

## 3 ALGORITHM FOR BILEVEL ERM

In this section, we consider the ERM bilevel problem, namely (BO) with (ERM), for which we denote the hyperobjective by $F_S$. Our algorithm is presented in Algorithm 1. We prove the following result:

**Theorem 3.1.** *Assume 2.5 and 2.6 hold, and that $\alpha \leq \ell\kappa^3 \min\{\frac{1}{2\kappa}, \frac{L_0^g}{L_0^f}, \frac{L_1^g}{L_1^f}, \frac{\Delta_F}{\ell\kappa}\}$. Then there is an assignment of parameters $\lambda \asymp \ell\kappa^3\alpha^{-1}$, $\sigma^2 \asymp \ell^2\kappa^2 T \log(T/\delta)\epsilon^{-2}n^{-2}$, $\eta \asymp \ell^{-1}\kappa^{-3}$, $T \asymp \Delta_F\ell\kappa^3\alpha^{-2}$, such that running Algorithm 1 satisfies $(\epsilon, \delta)$-DP, and returns $\mathbf{x}_{\mathrm{out}}$ such that with probability at least $1 - \gamma$ :*

$$\|\mathcal{G}_{F_S,\eta}(\mathbf{x}_{\mathrm{out}})\| \leq \alpha = \widetilde{\mathcal{O}}\left(K_1 \left(\frac{\sqrt{d_x}}{\epsilon n}\right)^{1/2} + K_2 \left(\frac{\sqrt{d_y}}{\epsilon n}\right)^{1/3}\right),$$

*where $K_1 = \mathcal{O}(\Delta_F^{1/4}\ell^{3/4}\kappa^{5/4})$, $K_2 = \mathcal{O}(\Delta_F^{1/6}\ell^{1/2}\kappa^{11/6})$.*

**Remark 3.2.** *Recall that when $\mathcal{X} = \mathbb{R}^{d_x}$, then $\mathcal{G}_{F_S,\eta}(\mathbf{x}_{\mathrm{out}}) = \nabla F_S(\mathbf{x}_{\mathrm{out}})$.*

---

[3]If $g(\mathbf{x}, \cdot; \xi)$ were Lipschitz over $\mathbb{R}^{d_y}$ for all $\xi \in \Xi$, then so would $g(\mathbf{x}, \cdot)$, contradicting strong convexity.

---

**Algorithm 1** DP Bilevel

---

1: **Input:** Initialization $(\mathbf{x}_0, \mathbf{y}_0) \in \mathcal{X} \times \mathcal{Y}$, privacy budget $(\epsilon, \delta)$, penalty $\lambda > 0$, noise level $\sigma^2 > 0$, step size $\eta > 0$, iteration budget $T \in \mathbb{N}$.
2: **for** $t = 0, \ldots, T - 1$ **do**
3:      Apply $\left( \frac{\epsilon}{\sqrt{18T}}, \frac{\delta}{3T} \right)$-DP-Loc-GD (Algorithm 2) to solve       ▷ Strongly-convex problems

$$\widetilde{\mathbf{y}}_t \approx \arg\min_{\mathbf{y}} g(\mathbf{x}_t, \mathbf{y})$$

$$\widetilde{\mathbf{y}}_t^\lambda \approx \arg\min_{\mathbf{y}} \left[ f(\mathbf{x}_t, \mathbf{y}) + \lambda \cdot g(\mathbf{x}_t, \mathbf{y}) \right]$$

4:      $\widetilde{\mathbf{g}}_t = \nabla_x f(\mathbf{x}_t, \widetilde{\mathbf{y}}_t^\lambda) + \lambda \left( \nabla_x g(\mathbf{x}_t, \widetilde{\mathbf{y}}_t^\lambda) - \nabla_x g(\mathbf{x}_t, \widetilde{\mathbf{y}}_t) \right) + \nu_t$, where $\nu_t \sim \mathcal{N}(\mathbf{0}, \sigma^2 I_{d_x})$
5:      $\mathbf{x}_{t+1} = \arg\min_{\mathbf{u} \in \mathcal{X}} \left\{ \langle \widetilde{\mathbf{g}}_t, \mathbf{u} \rangle + \frac{1}{2\eta} \|\mathbf{u} - \mathbf{x}_t\|^2 \right\}$      ▷ If $\mathcal{X} = \mathbb{R}^{d_x}$, then $\mathbf{x}_{t+1} = \mathbf{x}_t - \eta \widetilde{\mathbf{g}}_t$
6: **end for**
7: $t_{\text{out}} = \arg\min_{t \in \{0, \ldots, T-1\}} \|\mathbf{x}_{t+1} - \mathbf{x}_t\|$.
8: **Output:** $\mathbf{x}_{t_{\text{out}}}$.

---

**Algorithm 2** DP-Loc-GD

---

1: **Input:** Objective $h : \mathbb{R}^{d_y} \to \mathbb{R}$, initialization $\mathbf{y}_0 \in \mathcal{Y}$, privacy budget $(\epsilon', \delta')$, number of rounds $M \in \mathbb{N}$, noise level $\sigma_{\text{GD}}^2 > 0$, step sizes $(\eta_t)_{t=0}^{T-1}$, iteration budget $T_{\text{GD}} \in \mathbb{N}$, radii $(R_m)_{m=0}^{M-1} > 0$.
2: $\mathbf{y}_0^0 = \mathbf{y}_0$
3: **for** $m = 0, \ldots, M - 1$ **do**
4:      **for** $t = 0, \ldots, T_{\text{GD}} - 1$ **do**
5:          $\mathbf{y}_{t+1}^m = \text{Proj}_{\mathbb{B}(\mathbf{y}_0^m, R_m)} \left[ \mathbf{y}_t^m - \eta_t \left( \nabla h(\mathbf{y}_t^m) + \nu_t \right) \right]$, where $\nu_t^m \sim \mathcal{N}(\mathbf{0}, \sigma_{\text{GD}}^2 I_{d_y})$
6:      **end for**
7:      $\mathbf{y}_0^{m+1} = \frac{1}{T} \sum_{t=0}^{T-1} \mathbf{y}_t^m$
8: **end for**
9: **Output:** $\mathbf{y}_{\text{out}} = \mathbf{y}_0^M$.

---

### 3.1 ANALYSIS OVERVIEW

In this section, we will go over the main ideas that appear in the proof of Theorem 3.1, all which are provided in full detail in Appendix A. We start by introducing some useful notation: Given $\lambda > 0$, we denote the penalty function

$$\mathcal{L}_\lambda(\mathbf{x}, \mathbf{y}) := f(\mathbf{x}, \mathbf{y}) + \lambda \left[ g(\mathbf{x}, \mathbf{y}) - g(\mathbf{x}, \mathbf{y}^*(\mathbf{x})) \right] ,$$

and further denote

$$\mathcal{L}_\lambda^*(\mathbf{x}) := \mathcal{L}_\lambda(\mathbf{x}, \mathbf{y}^\lambda(\mathbf{x})) , \quad \mathbf{y}^\lambda(\mathbf{x}) := \arg\min_{\mathbf{y}} \mathcal{L}_\lambda(\mathbf{x}, \mathbf{y}) .$$

The starting point of our analysis is the following result, underlying the previously discussed recent advancements in (non-private) first-order bilevel optimization:

**Lemma 3.3** (Kwon et al. 2023; 2024; Chen et al. 2024). *For $\lambda \geq 2L_1^f/\mu_g$, the following hold:*

     *a. $\|\mathcal{L}_\lambda^* - F\|_\infty = \mathcal{O}(\ell\kappa/\lambda)$.*

     *b. $\|\nabla\mathcal{L}_\lambda^* - \nabla F\|_\infty = \mathcal{O}(\ell\kappa^3/\lambda)$.*

     *c. $\mathcal{L}_\lambda^*$ is $\mathcal{O}(\ell\kappa^3)$-smooth (independently of $\lambda$).*

In other words, the lemma shows that for sufficiently large penalty $\lambda$, $\mathcal{L}_\lambda^*$ is a smooth approximation of the hyperobjective $F$, and that it suffices to minimize the gradient norm of $\mathcal{L}_\lambda^*$ in order to get a hypergradient guarantee in terms of $\nabla F$. Moreover, note that $\nabla\mathcal{L}_\lambda^*$ can be computed entirely in a

first-order fashion, since by construction $\mathcal{L}_\lambda^*(\mathbf{x}) = \arg\min_\mathbf{y} \mathcal{L}_\lambda(\mathbf{x}, \mathbf{y})$, and therefore it holds that

$$\nabla \mathcal{L}_\lambda^*(\mathbf{x}) = \nabla_x \mathcal{L}_\lambda^*(\mathbf{x}, \mathbf{y}^\lambda(\mathbf{x})) + \nabla_x \mathbf{y}^\lambda(\mathbf{x})^\top \underbrace{\nabla_y \mathcal{L}_\lambda(\mathbf{x}, \mathbf{y}^\lambda(\mathbf{x}))}_{=\mathbf{0}}$$

$$= \nabla_x f(\mathbf{x}, \mathbf{y}^\lambda(\mathbf{x})) + \lambda \left( \nabla_x g(\mathbf{x}, \mathbf{y}^\lambda(\mathbf{x})) - \nabla_x g(\mathbf{x}, \mathbf{y}^*(\mathbf{x})) \right) \ . \tag{3}$$

This observation raises a subtle privacy issue: Since $\mathbf{y}^*(\mathbf{x}), \mathbf{y}^\lambda(\mathbf{x})$ are required in order to compute the gradient $\nabla \mathcal{L}_\lambda^*(\mathbf{x})$, and are defined as the minimizers of $g(\mathbf{x}, \cdot), \mathcal{L}_\lambda(\mathbf{x}, \cdot)$ which are data-dependent, we cannot simply compute them up to arbitraily small accuracy under the DP constraint. In other words, even deciding *where* to invoke the gradient oracles, can already leak user information, hence breaking privacy before the gradients are even computed. We therefore must resort to approximating them using an auxiliary private method, for which we use DP-Loc(alized)-GD (Algorithm 2).[4] We then crucially rely on the fact that $g(\mathbf{x}, \cdot) \ \mathcal{L}_\lambda(\mathbf{x}, \cdot)$ are both strongly-convex, which implies that optimizing them produces $\widetilde{\mathbf{y}}_t, \widetilde{\mathbf{y}}_t^\lambda$ such that the *distances* to the minimizers, namely $\|\widetilde{\mathbf{y}}_t - \mathbf{y}^*(\mathbf{x}_t)\|, \|\widetilde{\mathbf{y}}_t^\lambda - \mathbf{y}^\lambda(\mathbf{x}_t)\|$, are small. The distance bound is key, as Eq. (3) allows using the smoothness of $f, g$ to translate the distance bounds into an inexact (i.e. biased) gradient oracle for $\nabla \mathcal{L}_\lambda^*(\mathbf{x}_t)$, computed at the private points $\widetilde{\mathbf{y}}_t, \widetilde{\mathbf{y}}_t^\lambda$. Using this analysis we obtain the following guarantee:

**Lemma 3.4.** *If* $\lambda \geq \max\{\frac{2L_1^g}{\mu_g}, \frac{L_0^f}{L_0^g}, \frac{L_1^f}{L_1^g}\}$, *then there is* $\beta = \widetilde{\mathcal{O}}\left( \frac{\lambda \ell \kappa \sqrt{d_y T}}{\epsilon n} \right)$ *such that with probability at least* $1 - \gamma$, *for all* $t \in \{0, \dots, T-1\}$ :

$$\left\| \nabla \mathcal{L}_\lambda^*(\mathbf{x}_t) - \left[ \nabla_x f(\mathbf{x}_t, \widetilde{\mathbf{y}}_t^\lambda) + \lambda \left( \nabla_x g(\mathbf{x}_t, \widetilde{\mathbf{y}}_t^\lambda) - \nabla_x g(\mathbf{x}_t, \widetilde{\mathbf{y}}_t) \right) \right] \right\| \leq \beta \ .$$

Having constructed an inexact gradient oracle, we can privatize its response using the standard Gaussian mechanism. Recalling that the noise variance required to ensure privacy is tied to the component Lipschitz constants, we note that $\mathcal{L}_\lambda^*(\mathbf{x})$ decomposes as the finite-sum $\mathcal{L}_\lambda^*(\mathbf{x}) = \frac{1}{n} \sum_{i=1}^n \mathcal{L}_{\lambda,i}^*(\mathbf{x})$, where

$$\mathcal{L}_{\lambda,i}^*(\mathbf{x}) := f(\mathbf{x}, \mathbf{y}^\lambda(\mathbf{x}); \xi_i) + \lambda \left[ g(\mathbf{x}, \mathbf{y}^\lambda(\mathbf{x}); \xi_i) - g(\mathbf{x}, \mathbf{y}^*(\mathbf{x}); \xi_i) \right] \ .$$

At first glance, a naive application of the chain rule and the triangle inequality would bound the Lipschitz constant of $\mathcal{L}_{\lambda,i}^*$ by approximately $\mathrm{Lip}(\mathbf{y}^*)(L_0^f + \lambda L_0^g) \lesssim \lambda \mathrm{Lip}(\mathbf{y}^*) L_0^g$, where $\mathrm{Lip}(\mathbf{y}^*)$ is the Lipschitz constant of $\mathbf{y}^*(\mathbf{x}) : \mathbb{R}^{d_x} \to \mathbb{R}^{d_y}$. Unfortunately, this bound grows with the penalty parameter $\lambda$, which will eventually be set large, and in particular, will grow with the dataset size $n$. We therefore derive the following lemma, showing that applying a more nuanced analysis allows obtaining a significantly tighter Lipschitz bound, independent of $\lambda$ :

**Lemma 3.5.** $\mathcal{L}_{\lambda,i}^*$ *is* $\mathcal{O}(\ell \kappa)$-*Lipschitz.*

Finally, having constructed a private inexact stochastic oracle response for the smooth approximation $\mathcal{L}_\lambda^*$, we analyze an outer loop (Line 5 of Algorithm 1), showing that is provably robust to inexact and noisy gradients. We then employ a stopping criteria which makes use of the already-privatized iterates, thus avoiding the need of additional noise in choosing the smallest gradient. In particular, we show that the corresponding process gets to a point with small (projected-)gradient norm, as stated below:

**Proposition 3.6.** *Suppose* $h : \mathbb{R}^d \to \mathbb{R}$ *is* $L$-*smooth, that* $\|\widetilde{\nabla} h(\cdot) - \nabla h(\cdot)\| \leq \beta$, *and consider the following update rule with* $\eta = \frac{1}{2L}$ :

$$\mathbf{x}_{t+1} = \arg\min_{\mathbf{u} \in \mathcal{X}} \left\{ \left\langle \widetilde{\nabla} h(\mathbf{x}_t) + \nu_t, \mathbf{u} \right\rangle + \frac{1}{2\eta} \|\mathbf{x}_t - \mathbf{u}\|^2 \right\} \ , \quad \nu_t \sim \mathcal{N}(0, \sigma^2 I) \ ,$$

*with the output rule* $\mathbf{x}_{\mathrm{out}} := \mathbf{x}_{t_{\mathrm{out}}}$, $t_{\mathrm{out}} := \arg\min_{t \in \{0, \dots, T-1\}} \|\mathbf{x}_{t+1} - \mathbf{x}_t\|$. *If* $\alpha > 0$ *is such that* $\alpha \geq C \max\{\beta, \sigma \sqrt{d \log(T/\gamma)}\}$ *for a sufficiently large absolute constant* $C > 0$, *then with probability at least* $1 - \gamma$ : $\|\mathcal{G}_{h,\eta}(\mathbf{x}_{\mathrm{out}})\| \leq \alpha$ *for* $T = \mathcal{O}\left( \frac{L(h(\mathbf{x}_0) - \inf h)}{\alpha^2} \right)$.

Overall, applying Proposition 3.6 to $h = \mathcal{L}_\lambda^*$, we see that the (projected-)gradient norm can be as small as $\max\{\beta, \sigma \sqrt{d_x}\}$, up to logarithmic terms. Accounting for the smallest possible inexactness $\beta$ and noise addition $\sigma$ that ensure the the inner and outer loops, respectively, are both sufficiently private, we conclude the proof of Theorem 3.1; the full details appear in Appendix A.

---

[4]We can replace the inner solver by any DP method that guarantees with high probability the optimal rate for strongly-convex objectives, as we further discuss in Appendix B.

# 4 MINI-BATCH ALGORITHM FOR BILEVEL ERM

---

**Algorithm 3** Mini-batch DP Bilevel

---

1: **Input:** Initialization $(\mathbf{x}_0, \mathbf{y}_0) \in \mathcal{X} \times \mathcal{Y}$, privacy budget $(\epsilon, \delta)$, penalty $\lambda > 0$, noise level $\sigma^2 > 0$, step size $\eta > 0$, iteration budget $T \in \mathbb{N}$, batch sizes $b_{\text{in}}, b_{\text{out}} \in \mathbb{N}$.

2: **for** $t = 0, \ldots, T - 1$ **do**

3:     Apply $(\frac{\epsilon}{\sqrt{18T}}, \frac{\delta}{3T})$-DP-Loc-SGD (Algorithm 4) to solve    ▷ Strongly-convex problems

$$\widetilde{\mathbf{y}}_t \approx \arg\min_{\mathbf{y}} g(\mathbf{x}_t, \mathbf{y})$$

$$\widetilde{\mathbf{y}}_t^\lambda \approx \arg\min_{\mathbf{y}} [f(\mathbf{x}_t, \mathbf{y}) + \lambda \cdot g(\mathbf{x}_t, \mathbf{y})]$$

4:     $\widetilde{\mathbf{g}}_t = \nabla_x f(\mathbf{x}_t, \widetilde{\mathbf{y}}_t^\lambda; B_t) + \lambda \left(\nabla_x g(\mathbf{x}_t, \widetilde{\mathbf{y}}_t^\lambda; B_t) - \nabla_x g(\mathbf{x}_t, \widetilde{\mathbf{y}}_t; B_t)\right) + \nu_t, \ B_t \sim S^{b_{\text{out}}}, \ \nu_t \sim \mathcal{N}(\mathbf{0}, \sigma^2 I_{d_x})$

5:     $\mathbf{x}_{t+1} = \arg\min_{\mathbf{u} \in \mathcal{X}} \left\{ \langle \widetilde{\mathbf{g}}_t, \mathbf{u} \rangle + \frac{1}{2\eta} \|\mathbf{u} - \mathbf{x}_t\|^2 \right\}$    ▷ If $\mathcal{X} = \mathbb{R}^{d_x}$, then $\mathbf{x}_{t+1} = \mathbf{x}_t - \eta \widetilde{\mathbf{g}}_t$

6: **end for**

7: $t_{\text{out}} = \arg\min_{t \in \{0, \ldots, T-1\}} \|\mathbf{x}_{t+1} - \mathbf{x}_t\|$.

8: **Output:** $\mathbf{x}_{t_{\text{out}}}$.

---

**Algorithm 4** DP-Loc-SGD

---

1: **Input:** Objective $h : \mathbb{R}^{d_y} \times \Xi \to \mathbb{R}$, initialization $\mathbf{y}_0 \in \mathcal{Y}$, privacy budget $(\epsilon', \delta')$, batch size $b_{\text{in}} \in \mathbb{N}$, number of rounds $M \in \mathbb{N}$, noise level $\sigma_{\text{SGD}}^2 > 0$, step sizes $(\eta_t)_{t=0}^{T-1}$, iteration budget $T_{\text{SGD}} \in \mathbb{N}$, radii $(R_m)_{m=0}^{M-1} > 0$.

2: $\mathbf{y}_0^0 = \mathbf{y}_0$

3: **for** $m = 0, \ldots, M - 1$ **do**

4:     **for** $t = 0, \ldots, T_{\text{SGD}} - 1$ **do**

5:         $\mathbf{y}_{t+1}^m = \text{Proj}_{\mathbb{B}(\mathbf{y}_0^m, R_m)} [\mathbf{y}_t^m - \eta_t (\nabla h(\mathbf{y}_t^m; B_t) + \nu_t)], \quad B_t^m \sim S^{b_{\text{in}}}, \ \nu_t^m \sim \mathcal{N}(\mathbf{0}, \sigma_{\text{SGD}}^2 I_{d_y})$

6:     **end for**

7:     $\mathbf{y}_0^{m+1} = \frac{1}{T} \sum_{t=0}^{T-1} \mathbf{y}_t^m$

8: **end for**

9: **Output:** $\mathbf{y}_{\text{out}} = \mathbf{y}_0^M$.

---

In this section, we consider once again the ERM bilevel problem, (BO) with (ERM), and provide Algorithm 3, which is a mini-batch variant of the ERM algorithm discussed in the previous section. Given a mini-batch $B \subseteq S = \{\xi_1, \ldots, \xi_n\}$ and a function $h : \mathbb{R}^d \times \Xi \to \mathbb{R}$, we let $\nabla h(\mathbf{z}; B) = \frac{1}{|B|} \sum_{\xi_i \in B} \nabla h(\mathbf{z}; \xi_i)$ denote the mini-batch gradient. We prove the following result:

**Theorem 4.1.** *Assume 2.5 and 2.6 hold, and that $\alpha \leq \ell \kappa^3 \min\{\frac{1}{2\kappa}, \frac{L_0^g}{L_0^f}, \frac{L_1^g}{L_1^f}, \frac{\Delta_F}{\ell \kappa}\}$. Then running Algorithm 3 with assigned parameters as in Theorem 3.1 and any batch sizes $b_{\text{in}}, b_{\text{out}} \in [n]$, satisfies $(\epsilon, \delta)$-DP and returns $\mathbf{x}_{\text{out}}$ such that with probability at least $1 - \gamma$ :*

$$\|\mathcal{G}_{F_S, \eta}(\mathbf{x}_{\text{out}})\| \leq \alpha = \widetilde{\mathcal{O}}\left( K_1 \left(\frac{\sqrt{d_x}}{\epsilon n}\right)^{1/2} + K_2 \left(\frac{\sqrt{d_y}}{\epsilon n}\right)^{1/3} + K_3 \cdot \frac{1}{b_{\text{out}}} \right),$$

*where $K_1 = \mathcal{O}(\Delta_F^{1/4} \ell^{3/4} \kappa^{5/4})$, $K_2 = \mathcal{O}(\Delta_F^{1/6} \ell^{1/2} \kappa^{11/6})$, $K_3 = \mathcal{O}(\ell \kappa)$.*

**Remark 4.2** (Outer batch size dependence)**.** *Algorithm 3 ensures privacy for any batch sizes, yet notably, the guaranteed gradient norm bound does not go to zero (as $n \to \infty$) for constant outer-batch size. The same phenomenon also holds for for "single"-level constrained nonconvex optimization, as noted by Ghadimi et al. (2016) (specifically, see Corollary 4 and related discussion). Accordingly, the inner-batch size $b_{\text{in}}$ can be set whatsoever, while setting $b_{\text{out}} = \mathcal{O}(\max\{(\epsilon n/\sqrt{d_x})^{1/2}, (\epsilon n/\sqrt{d_y})^{1/3}\}) \ll n$ recovers the full-batch rate. More generally, from a worst-case perspective, one should set $b_{\text{out}} = \omega_n(1)$ to grow with the sample (resulting in*

*$\lim_{n\to\infty} \|\mathcal{G}(\mathbf{x}_{\mathrm{out}})\| = 0$). It is interesting to note that the additional $1/b_{\mathrm{out}}$ term shows up in the analysis only as an upper bound on the sub-Gaussian norm of the mini-batch gradient estimator. Thus, in applications for which some (possibly constant) batch size results in reasonably accurate gradients, the result above should hold with the outer mini-batch gradient's standard deviation replacing $1/b_{\mathrm{out}}$, which is to be expected anyhow for high probability guarantees.*

The difference between Algorithm 3 and Algorithm 1, is that both the inner and outer loops sample mini-batch gradients. The inner loop guarantee is the same regardless of the inner-batch size $b_{\mathrm{in}}$, since for strongly-convex objectives it is possible to prove the same convergence rate guarantee for DP optimization in any case (as further discussed in Appendix B). As to the outer loop (Line 5), we apply standard concentration bounds to argue about the quality of the gradient estimates — hence the additive $1/b_{\mathrm{out}}$ factor — and rely on our analysis of the outer loop with inexact gradients (which are now even less exact due to sampling stochasticity). We remark that compared to the classic analysis of Ghadimi et al. (2016) for constrained nonconvex optimization, we derive high probability bounds without requiring several re-runs of the algorithm. We further remark that we analyze mini-batch sampling with replacement for simplicity, though the same guarantees (up to constants) can be derived for sampling at each time step without replacement, at the cost of a more involved analysis.

## 5 GENERALIZING FROM ERM TO POPULATION LOSS

In this section, we move to consider stochastic (population) objectives, the problem (BO) with (Pop). We denote the population hyperobjective by $F_{\mathcal{P}}$, and as before $F_S$ denotes the empirical objective, where $S \sim \mathcal{P}^n$. We prove the following result:

**Theorem 5.1.** *Under Assumptions 2.5 and 2.6, if the preconditions of Theorem 3.1 hold, then Algorithm 1 is $(\epsilon, \delta)$-DP, and returns $\mathbf{x}_{\mathrm{out}}$ such that with probability at least $1 - \gamma$ :*

$$\|\mathcal{G}_{F_{\mathcal{P}}, \eta}(\mathbf{x}_{\mathrm{out}})\| \leq \alpha = \widetilde{\mathcal{O}}\left(K_1 \left(\frac{\sqrt{d_x}}{\epsilon n}\right)^{1/2} + K_2 \left(\frac{\sqrt{d_y}}{\epsilon n}\right)^{1/3} + K_3 \left(\frac{d_x}{n}\right)^{1/2}\right),$$

*where $K_1 = \mathcal{O}(\Delta_F^{1/4} \ell^{1/4} \kappa^{5/4})$, $K_2 = \mathcal{O}(\Delta_F^{1/6} \ell^{1/2} \kappa^{11/6})$, $K_3 = \mathcal{O}(\ell\kappa)$. Similarly, if the preconditions of Theorem 4.1 hold, then for any batch sizes $b_{\mathrm{in}}, b_{\mathrm{out}} \in [n]$, Algorithm 3 is $(\epsilon, \delta)$-DP, and returns $\mathbf{x}_{\mathrm{out}}$ such that with probability at least $1 - \gamma$ :*

$$\|\mathcal{G}_{F_{\mathcal{P}}, \eta}(\mathbf{x}_{\mathrm{out}})\| \leq \alpha = \widetilde{\mathcal{O}}\left(K_1 \left(\frac{\sqrt{d_x}}{\epsilon n}\right)^{1/2} + K_2 \left(\frac{\sqrt{d_y}}{\epsilon n}\right)^{1/3} + K_3 \cdot \frac{1}{b_{\mathrm{out}}} + K_3 \left(\frac{d_x}{n}\right)^{1/2}\right).$$

The proof of Theorem 5.1 relies on arguing that the hyperobjective is Lipschitz, and applying a uniform convergence bound for bounded gradients, which further implies uniform convergence of projected gradients by Lemma 2.4.

## 6 DISCUSSION

In this paper, we studied differentially-private bilevel optimization, and proposed the first algorithms to solve this problem that enable any desired privacy guarantee, while also requiring only gradient queries. Our provided guarantees hold both for constrained and unconstrained settings, cover empirical and population losses alike, and account for mini-batched gradients.

Our work leaves open several directions for future research. First, it is likely that the rate derived in this work can be improved. Specifically, for "single"-level DP nonconvex optimization, Arora et al. (2023) showed that incorporating variance reduction leads to gradient bounds that decay faster with the sample size. Applying this for DP bilevel optimization as the outer loop would require, according to our analysis, to evaluate the cost of inexact gradients in variance-reduced methods, which we leave for future work.

Another open direction is understanding whether mini-batch algorithms can avoid the additive $1/b_{\mathrm{out}}$ factor in the *unconstrained* case $\mathcal{X} = \mathbb{R}^{d_x}$. As previously discussed, for constrained problems, even single-level nonconvex algorithms suffer from this batch dependence (Ghadimi et al.,

2016). Nonetheless, for unconstrained problems, Ghadimi & Lan (2013) showed that setting a smaller stepsize, roughly on the order of $\alpha^2/\sigma^2$, converges to a point with gradient bounded by $\alpha$ after $\mathcal{O}(\alpha^{-4})$ steps, even for $b_{\text{out}} = 1$. Applying this to DP bilevel unconstrained optimization, would require analyzing SGD under biased gradients, and accounting for the larger privacy loss due to the slower convergence rate (compared to $\mathcal{O}(\alpha^{-2})$ in our case), both of which seem feasible.

Lastly, an important direction is of course empirical validation of our proposed methods. At a high level, our methods are privatized variants of the first-order penalty approach for bilevel optimization, which has been substantially scaled up following initial theoretically-focused works, confirming this paradigm as highly effective in some large scale non-private applications (Pan et al., 2024). While our analysis provides conservative (worst-case) estimates for the convergence rate under privatization of both the upper and lower level problems, it would be interesting to explore the actual cost of privatization seen in practice for these problems. As this work is a theoretically focused, we leave this for future research.

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

# A    PROOFS

Throughout the proof section, we abbreviate $f_i(\cdot) = f(\cdot\,;\xi_i),\ g_i(\cdot) = g(\cdot\,;\xi_i),\ F = F_S$.

## A.1    PROOF OF LEMMA 3.4

Note that the two sub-problems solved by Line 3 of Algorithm 1 are strongly-convex and admit Lipschitz components over $\mathcal{Y}$; $g(\mathbf{x},\cdot)$ by assumption, and $f + \lambda g$ by combining this with the smoothness/Lipschitzness of $f$, as follows:

**Lemma A.1.** *If $\lambda \geq \max\{\frac{2L_1^g}{\mu_g}, \frac{L_0^f}{L_0^g}\}$ then for all $\mathbf{x} \in \mathcal{X}$ : $f(\mathbf{x},\cdot) + \lambda g(\mathbf{x},\cdot)$ is $\frac{\lambda\mu_g}{2}$ strongly-convex, and moreover for all $i \in [n]$ : $f_i(\mathbf{x},\cdot) + \lambda g_i(\mathbf{x},\cdot)$ is $2\lambda L_0^g$-Lipschitz.*

We therefore invoke the following guarantee, which provides the optimal result for strongly-convex DP ERM via DP-Loc-GD (Algorithm 2).

**Theorem A.2.** *Suppose that $h : \mathbb{R}^{d_y} \to \mathbb{R}$ is a $\mu$-strongly-convex function of the form $h(\mathbf{y}) = \frac{1}{n}\sum_{i=1}^n h(\mathbf{y}, \xi_i)$ where $h(\cdot, \xi_i)$ is $L$-Lipschitz for all $i \in [n]$. Suppose $\arg\min h =: \mathbf{y}^* \in \mathbb{B}(\mathbf{y}_0, R_0)$ and that $n \geq \frac{LR_0^{2/\log(d_y)}}{\mu\epsilon'}$. Then there is an assignment of parameters $M = \log_2\log(\frac{\mu\epsilon' n}{L})$, $\sigma_{\mathrm{GD}}^2 = \widetilde{\mathcal{O}}(L^2/\epsilon'^2)$, $\eta_t = \frac{1}{\mu(t+1)}$, $T_{\mathrm{GD}} = n^2$, $R_m = \widetilde{\Theta}\left(\sqrt{\frac{R_{m-1}L}{\mu\epsilon' n}} + \frac{L\sqrt{d_y}}{\mu\epsilon' n}\right)$ such that running Algorithm 2 satisfies $(\epsilon', \delta')$-DP, and outputs $\mathbf{y}_{\mathrm{out}}$ such that $\|\mathbf{y}_{\mathrm{out}} - \mathbf{y}^*\| = \widetilde{\mathcal{O}}\left(\frac{L\sqrt{d_y}}{\mu n \epsilon'}\right)$ with probability at least $1 - \gamma$.*

Although the rate in Theorem A.2 appears in prior works such as (Bassily et al., 2014; Feldman et al., 2020), it is typically manifested through a bound in expectation (and in terms of function value) as opposed to with high probability, required for our purpose. We therefore, for the sake of completeness, provide a self-contained proof of Theorem A.2 in Appendix B.

Applied to the functions $g(\mathbf{x}_t, \cdot)$ and $f(\mathbf{x}_t, \cdot) + \lambda g(\mathbf{x}_t, \cdot)$, and invoking Lemma A.1, yields the following.

**Corollary A.3.** *If $\lambda \geq \max\{\frac{2L_1^g}{\mu_g}, \frac{L_0^f}{L_0^g}\}$, then $\widetilde{\mathbf{y}}_t$ and $\widetilde{\mathbf{y}}_t^\lambda$ (as appear in Line 3 of Algorithm 1) satisfy with probability at least $1 - \gamma$ :*

$$\max\left\{\|\widetilde{\mathbf{y}}_t - \mathbf{y}^*(\mathbf{x}_t)\|,\, \|\widetilde{\mathbf{y}}_t^\lambda - \mathbf{y}^\lambda(\mathbf{x}_t)\|\right\} = \widetilde{\mathcal{O}}\left(\frac{L_0^g\sqrt{d_y T}}{\epsilon\mu_g n}\right)\,.$$

We are now ready to prove the main proposition of this section, which we restate below:

**Lemma 3.4.** *If $\lambda \geq \max\{\frac{2L_1^g}{\mu_g}, \frac{L_0^f}{L_0^g}, \frac{L_1^f}{L_1^g}\}$, then there is $\beta = \widetilde{\mathcal{O}}\left(\frac{\lambda\ell\kappa\sqrt{d_y T}}{\epsilon n}\right)$ such that with probability at least $1 - \gamma$, for all $t \in \{0, \ldots, T-1\}$ :*

$$\left\|\nabla\mathcal{L}_\lambda^*(\mathbf{x}_t) - \left[\nabla_x f(\mathbf{x}_t, \widetilde{\mathbf{y}}_t^\lambda) + \lambda\left(\nabla_x g(\mathbf{x}_t, \widetilde{\mathbf{y}}_t^\lambda) - \nabla_x g(\mathbf{x}_t, \widetilde{\mathbf{y}}_t)\right)\right]\right\| \leq \beta\,.$$

*Proof of Lemma 3.4.* As in Eq. (3), we note that by construction $\mathcal{L}_\lambda^*(\mathbf{x}) = \arg\min_{\mathbf{y}} \mathcal{L}_\lambda(\mathbf{x}, \mathbf{y})$, therefore it holds that

$$\nabla\mathcal{L}_\lambda^*(\mathbf{x}) = \nabla_x f(\mathbf{x}, \mathbf{y}^\lambda(\mathbf{x})) + \lambda\left(\nabla_x g(\mathbf{x}, \mathbf{y}^\lambda(\mathbf{x})) - \nabla_x g(\mathbf{x}, \mathbf{y}^*(\mathbf{x}))\right)\,.$$

Denoting $\mathbf{g}_t = \nabla_x f(\mathbf{x}_t, \widetilde{\mathbf{y}}_t^\lambda) + \lambda\left(\nabla_x g(\mathbf{x}_t, \widetilde{\mathbf{y}}_t^\lambda) - \nabla_x g(\mathbf{x}_t, \widetilde{\mathbf{y}}_t)\right)$, by the smoothness of $f$ and $g$ we see that

$$\|\mathbf{g}_t - \nabla\mathcal{L}_\lambda^*(\mathbf{x}_t)\| \leq L_1^f \left\|\widetilde{\mathbf{y}}_t^\lambda - \mathbf{y}^\lambda(\mathbf{x}_t)\right\| + \lambda L_1^g \left\|\widetilde{\mathbf{y}}_t^\lambda - \mathbf{y}^\lambda(\mathbf{x}_t)\right\| + \lambda L_1^g \left\|\widetilde{\mathbf{y}}_t - \mathbf{y}^*(\mathbf{x}_t)\right\|\,.$$

Applying Corollary A.3 and union bounding over $T$, we can further bound the above as

$$\|\mathbf{g}_t - \nabla \mathcal{L}_\lambda^*(\mathbf{x}_t)\| = \widetilde{\mathcal{O}} \left( \frac{L_1^f L_0^g \sqrt{d_y T}}{\epsilon \mu_g n} + \frac{\lambda L_1^g L_0^g \sqrt{d_y T}}{\epsilon \mu_g n} + \frac{\lambda L_1^g L_0^g \sqrt{d_y T}}{\epsilon \mu_g n} \right)$$

$$= \widetilde{\mathcal{O}} \left( \frac{\lambda L_1^g L_0^g \sqrt{d_y T}}{\epsilon \mu_g n} \right)$$

$$= \widetilde{\mathcal{O}} \left( \frac{\lambda \ell \kappa \sqrt{d_y T}}{\epsilon n} \right) ,$$

where the second bound holds for $\lambda \geq \frac{L_1^f}{L_1^g}$ . $\qquad \square$

### A.2   PROOF OF LEMMA 3.5

We start by providing two lemmas, both of which borrow ideas that appeared in the smoothness analysis of Chen et al. (2024), and are proved here for completeness.

**Lemma A.4.** $\mathbf{y}^\lambda(\mathbf{x}) : \mathbb{R}^{d_x} \to \mathbb{R}^{d_y}$ is $\left( \frac{4L_1^g}{\mu_g} \right)$-Lipschitz.

*Proof of Lemma A.4.* Differentiating $\nabla_y \mathcal{L}_\lambda(\mathbf{x}, \mathbf{y}^\lambda(\mathbf{x})) = \mathbf{0}$ with respect the first argument gives

$$\nabla_{xy}^2 \mathcal{L}_\lambda(\mathbf{x}, \mathbf{y}^\lambda(\mathbf{x})) + \nabla \mathbf{y}^\lambda(\mathbf{x}) \cdot \nabla_{yy} \mathcal{L}_\lambda(\mathbf{x}, \mathbf{y}^\lambda(\mathbf{x})) = \mathbf{0} ,$$

hence

$$\nabla \mathbf{y}^\lambda(\mathbf{x}) = -\nabla_{xy}^2 \mathcal{L}_\lambda(\mathbf{x}, \mathbf{y}^\lambda(\mathbf{x})) \cdot \left[ \nabla_{yy} \mathcal{L}_\lambda(\mathbf{x}, \mathbf{y}^\lambda(\mathbf{x})) \right]^{-1} .$$

Noting that $\nabla_{xy}^2 \mathcal{L}_\lambda \preceq 2\lambda L_1^g$ and $\nabla_{yy}^2 \mathcal{L}_\lambda \succeq \lambda \mu_g/2$ everywhere, hence $[\nabla_{yy}^2 \mathcal{L}_\lambda]^{-1} \preceq 2/\lambda\mu_g$ we get that

$$\left\| \nabla \mathbf{y}^\lambda(\mathbf{x}) \right\| \leq 2\lambda L_1^g \cdot \frac{2}{\lambda \mu_g} = \frac{4L_1^g}{\mu_g} .$$

$\qquad \square$

**Lemma A.5.** *For all* $\mathbf{x} \in \mathcal{X}$: $\left\| \mathbf{y}^\lambda(\mathbf{x}) - \mathbf{y}^*(\mathbf{x}) \right\| \leq \frac{L_0^f}{\lambda \mu_g}$.

*Proof of Lemma A.5.* First, note that by definition of $\mathbf{y}^\lambda(\mathbf{x})$ it holds that

$$\mathbf{0} = \nabla_y \mathcal{L}_\lambda(\mathbf{x}, \mathbf{y}^\lambda(\mathbf{x})) = \nabla_y f(\mathbf{x}, \mathbf{y}^\lambda(\mathbf{x})) + \lambda \nabla_y g(\mathbf{x}, \mathbf{y}^\lambda(\mathbf{x})) ,$$

hence

$$\nabla_y g(\mathbf{x}, \mathbf{y}^\lambda(\mathbf{x})) = -\frac{1}{\lambda} \cdot \nabla_y f(\mathbf{x}, \mathbf{y}^\lambda(\mathbf{x})) ,$$

so in particular by the Lipschitz assumption on $f$ we see that

$$\left\| \nabla_y g(\mathbf{x}, \mathbf{y}^\lambda(\mathbf{x})) \right\| \leq \frac{L_0^f}{\lambda} .$$

By invoking the $\mu$-strong convexity of $g$ we further get

$$\left\| \mathbf{y}^\lambda(\mathbf{x}) - \mathbf{y}^*(\mathbf{x}) \right\| \leq \frac{1}{\mu} \left\| \nabla_y g(\mathbf{x}, \mathbf{y}^\lambda(\mathbf{x})) - \underbrace{\nabla_y g(\mathbf{x}, \mathbf{y}^*(\mathbf{x}))}_{=\mathbf{0}} \right\| \leq \frac{L_0^f}{\lambda \mu} .$$

$\qquad \square$

We are now ready to prove the main result of this section, restated below.

**Lemma 3.5.** $\mathcal{L}_{\lambda,i}^*$ is $\mathcal{O}(\ell\kappa)$-Lipschitz.

*Proof of Lemma 3.5.* For all $\mathbf{x} \in \mathcal{X}$ it holds that

$$\left\|\nabla \mathcal{L}_{\lambda,i}^*(\mathbf{x})\right\| = \left\|\nabla_x f_i(\mathbf{x}, \mathbf{y}^\lambda(\mathbf{x})) + \lambda \left[\nabla_x g_i(\mathbf{x}, \mathbf{y}^\lambda(\mathbf{x})) - \nabla_x g_i(\mathbf{x}, \mathbf{y}^*(\mathbf{x}))\right]\right\|$$

$$\leq \left\|\nabla_x f_i(\mathbf{x}, \mathbf{y}^\lambda(\mathbf{x}))\right\| + \lambda \left\|\nabla_x g_i(\mathbf{x}, \mathbf{y}^\lambda(\mathbf{x})) - \nabla_x g_i(\mathbf{x}, \mathbf{y}^*(\mathbf{x}))\right\| , \qquad (4)$$

thus we will bound each of the summands above.

For the first term, since $\mathbf{y}^\lambda$ is $\frac{4L_1^g}{\mu_g}$ Lipschitz according to Lemma A.4, and $f_i$ is $L_0^f$-Lipschitz by assumption, the chain rule yields the bound

$$\left\|\nabla_x f_i(\mathbf{x}, \mathbf{y}^\lambda(\mathbf{x}))\right\| \leq \frac{4L_1^g L_0^f}{\mu_g} \leq 4\ell\kappa . \qquad (5)$$

As to the second term, since $g_i$ is $L_1^g$-smooth, we use Lemma A.5 and get that

$$\lambda \left\|\nabla_x g_i(\mathbf{x}, \mathbf{y}^\lambda(\mathbf{x})) - \nabla_x g_i(\mathbf{x}, \mathbf{y}^*(\mathbf{x}))\right\| \leq \lambda L_1^g \left\|\mathbf{y}^\lambda(\mathbf{x}) - \mathbf{y}^*(\mathbf{x})\right\| \leq \frac{L_1^g L_0^f}{\mu_g} \leq \ell\kappa . \qquad (6)$$

Plugging Eqs. (5) and (6) into Eq. (4) completes the proof.

$\square$

## A.3 PROOF OF PROPOSITION 3.6

As $\nu_0, \ldots, \nu_{T-1} \overset{iid}{\sim} \mathcal{N}(0, \sigma^2 I)$, a standard Gaussian norm bound (cf. Vershynin 2018, Theorem 3.1.1) ensures that with probability at least $1 - \gamma$, for all $t \in \{0, 1, \ldots, T-1\}$ : $\|\nu_t\|^2 \lesssim d\sigma^2 \log(T/\gamma) \lesssim \frac{\alpha^2}{64}$. We therefore condition the rest of the proof on the highly probable event under which this uniform norm bound indeed holds. We continue by introducing some notation. We denote $\widetilde{\nabla}_t = \widetilde{\nabla} h(\mathbf{x}_t) + \nu_t$, and $\delta_t := \widetilde{\nabla}_t - \nabla h(\mathbf{x}_t)$. We further denote

$$\mathbf{x}_t^+ := \arg\min_{\mathbf{u} \in \mathcal{X}} \left\{\langle \nabla h(\mathbf{x}_t), \mathbf{u}\rangle + \frac{1}{2\eta} \|\mathbf{x}_t - \mathbf{u}\|^2\right\} ,$$

$$\mathcal{G}_t := \frac{1}{\eta}(\mathbf{x}_t - \mathbf{x}_t^+) ,$$

$$\rho_t := \frac{1}{\eta}(\mathbf{x}_t - \mathbf{x}_{t+1}) .$$

Note that by construction,

$$\mathcal{G}_t = \mathcal{G}_{h,\eta}(\mathbf{x}_t) := \frac{1}{\eta}\left(\mathbf{x}_t - \mathcal{P}_{\nabla h,\eta}(\mathbf{x}_t)\right) , \quad \mathcal{P}_{\nabla h,\eta}(\mathbf{x}_t) := \arg\min_{\mathbf{u} \in \mathcal{X}}\left[\langle \nabla h(\mathbf{x}_t), \mathbf{u}\rangle + \frac{1}{2\eta} \|\mathbf{u} - \mathbf{x}_t\|^2\right] ,$$

and that $\mathcal{G}_{t_{\text{out}}}$ is precisely the quantity we aim to bound. We start by proving some useful lemmas regarding the quantities defined above.

**Lemma A.6.** *Under the event that $\|\nu_t\|^2 \leq \frac{\alpha^2}{64}$ for all $t$, it holds that $\|\delta_t\| \leq \frac{\alpha}{4}$.*

*Proof.* By our assumptions on $\beta, \|\nu_t\|$, we get that

$$\|\delta_t\| \leq \|\widetilde{\nabla} h(\mathbf{x}_t) - \nabla h(\mathbf{x}_t)\| + \|\nu_t\| \leq \beta + \frac{\alpha}{8} \leq \frac{\alpha}{4} .$$

$\square$

**Lemma A.7.** *It holds that $\langle \widetilde{\nabla}_t, \rho_t\rangle \geq \|\rho_t\|^2$.*

*Proof.* By definition, $\mathbf{x}_{t+1} = \arg\min_{\mathbf{u} \in \mathcal{X}}\left\{\langle \widetilde{\nabla}_t, \mathbf{u}\rangle + \frac{1}{2\eta} \|\mathbf{x}_t - \mathbf{u}\|^2\right\}$. Hence, by the first-order optimality criterion, for any $\mathbf{u} \in \mathcal{X}$ :

$$\left\langle \widetilde{\nabla}_t + \frac{1}{\eta}(\mathbf{x}_{t+1} - \mathbf{x}_t), \mathbf{u} - \mathbf{x}_{t+1}\right\rangle \geq 0 .$$

In particular, setting $\mathbf{u} = \mathbf{x}_t$ yields

$$0 \leq \left\langle \widetilde{\nabla}_t + \frac{1}{\eta}(\mathbf{x}_{t+1} - \mathbf{x}_t), \mathbf{x}_t - \mathbf{x}_{t+1}\right\rangle = \left\langle \widetilde{\nabla}_t - \rho_t, \eta\rho_t\right\rangle = \eta\left(\left\langle \widetilde{\nabla}_t, \rho_t\right\rangle - \|\rho_t\|^2\right) ,$$

which proves the claim since $\eta > 0$.

$\square$

With the lemmas above in hand, we are now ready to prove Proposition 3.6. Note that by construction, the algorithm returns the index $t$ with minimal $\|\rho_t\|$. Further note that $\|\rho_t - \mathcal{G}_t\| \leq \|\delta_t\|$ by Lemma 2.4, thus

$$\|\mathcal{G}_t\| \leq \|\rho_t\| + \|\delta_t\| \leq \|\rho_t\| + \frac{\alpha}{4} , \tag{7}$$

where the last inequality is due to Lemma A.6, hence it suffices to bound $\|\rho_{t_{\text{out}}}\|$ (which is the quantity measured by the stopping criterion). To that end, by smoothness, we have for any $t \in \{0, 1, \ldots, T-2\}$ :

$$h(\mathbf{x}_{t+1}) \leq h(\mathbf{x}_t) + \langle \nabla h(\mathbf{x}_t), \mathbf{x}_{t+1} - \mathbf{x}_t \rangle + \frac{L}{2} \|\mathbf{x}_{t+1} - \mathbf{x}_t\|^2$$

$$= h(\mathbf{x}_t) - \eta \langle \nabla h(\mathbf{x}_t), \rho_t \rangle + \frac{L\eta^2}{2} \|\rho_t\|^2$$

$$= h(\mathbf{x}_t) - \eta \left\langle \widetilde{\nabla}_t, \rho_t \right\rangle + \frac{L\eta^2}{2} \|\rho_t\|^2 + \eta \langle \delta_t, \rho_t \rangle$$

$$\leq h(\mathbf{x}_t) - \eta \|\rho_t\|^2 + \frac{L\eta^2}{2} \|\rho_t\|^2 + \eta \|\delta_t\| \cdot \|\rho_t\| ,$$

where the last inequality followed by applying Lemma A.7 and Cauchy-Schwarz. Rearranging, and recalling that $\eta = \frac{1}{2L}$, hence $1 < 2 - L\eta$ and also $\frac{1}{\eta} = 2L$, we get that

$$\|\rho_t\|^2 - 2\|\delta_t\| \cdot \|\rho_t\| \leq (2 - L\eta) \|\rho_t\|^2 - 2\|\delta_t\| \cdot \|\rho_t\| \leq \frac{2(h(\mathbf{x}_t) - h(\mathbf{x}_{t+1}))}{\eta} = 4L(h(\mathbf{x}_t) - h(\mathbf{x}_{t+1})) .$$

Summing over $t \in \{0, 1 \ldots, T-1\}$, using the telescoping property of the right hand side, and dividing by $T$ gives that

$$\frac{1}{T} \sum_{t=0}^{T-1} \|\rho_t\| (\|\rho_t\| - 2\|\delta_t\|) \leq \frac{4L(h(\mathbf{x}_0) - \inf h)}{T} . \tag{8}$$

Note that if for some $t \in \{0, 1, \ldots, T-1\}$ : $\|\rho_t\| \leq \frac{3\alpha}{4}$ then we have proved our desired claim by Eq. (7) and the fact that $\|\rho_{t_{\text{out}}}\| = \min_t \|\rho_t\|$ by definition. On the other hand, assuming that $\|\rho_t\| > \frac{3\alpha}{4}$ for all $t$, invoking Lemma A.6, we see that $\|\rho_t\| - 2\|\delta_t\| \geq \|\rho_t\| - \frac{\alpha}{2} \geq \frac{1}{3} \|\rho_t\|$, which implies $\|\rho_t\| (\|\rho_t\| - 2\|\delta_t\|) \geq \frac{1}{3} \|\rho_t\|^2$. Combining this with Eq. (8) yields

$$\|\rho_{t_{\text{out}}}\|^2 = \min_{t \in \{0, 1, \ldots, T-1\}} \|\rho_t\|^2 \leq \frac{1}{T} \sum_{t=0}^{T-1} \|\rho_t\|^2 \leq \frac{12L(h(\mathbf{x}_0) - \inf h)}{T} ,$$

and the right side is bounded by $\frac{9\alpha^2}{16}$ for $T = \mathcal{O}\left(\frac{L(h(\mathbf{x}_0) - \inf h)}{\alpha^2}\right)$, finishing the proof by Eq. (7).

## A.4 PROOF OF THEOREM 3.1

We start by proving the privacy guarantee. Since $\mathcal{L}_{\lambda,i}^*$ is $\mathcal{O}(\ell\kappa)$-Lipschitz by Lemma 3.5, the sensitivity of $\nabla_x f(\mathbf{x}_t, \widetilde{\mathbf{y}}_t^\lambda) + \lambda \left(\nabla_x g(\mathbf{x}_t, \widetilde{\mathbf{y}}_t^\lambda) - \nabla_x g(\mathbf{x}_t, \widetilde{\mathbf{y}}_t)\right)$ is at most $\mathcal{O}(\ell\kappa)$. Hence, by setting $\sigma^2 = C \frac{\ell^2 \kappa^2 \log(T/\delta)T}{\epsilon^2 n^2}$ for a sufficiently large absolute constant $C > 0$, $\widetilde{\mathbf{g}}_t$ is $(\frac{\epsilon}{\sqrt{18T}}, \frac{\delta}{3(T+1)})$-DP. By basic composition, since each iteration also runs $(\frac{\epsilon}{\sqrt{18T}}, \frac{\delta}{3(T+1)})$-DP-Loc-GD twice, we see that each iteration of the algorithm is $(3 \cdot \frac{\epsilon}{\sqrt{18T}}, 3 \cdot \frac{\delta}{3(T+1)}) = (\frac{\epsilon}{\sqrt{2T}}, \frac{\delta}{(T+1)})$-DP. Noting that under our parameter assignment $\frac{\epsilon}{\sqrt{T}} \ll 1$, by advanced composition we get that throughout $T$ iterations, the algorithm is overall $(\epsilon, \delta)$-DP as claimed.

We turn to analyze the utility of the algorithm. It holds that

$$\|\mathcal{G}_{F,\eta}(\mathbf{x}_{t_{\text{out}}})\| \leq \left\|\mathcal{G}_{F,\eta}(\mathbf{x}_{t_{\text{out}}}) - \mathcal{G}_{\mathcal{L}_\lambda^*,\eta}(\mathbf{x})\right\| + \left\|\mathcal{G}_{\mathcal{L}_\lambda^*,\eta}(\mathbf{x}_{t_{\text{out}}})\right\|$$

$$\leq \|\nabla F(\mathbf{x}_{t_{\text{out}}}) - \nabla \mathcal{L}_\lambda^*(\mathbf{x})\| + \|\mathcal{G}_{\mathcal{L}_\lambda^*,\eta}(\mathbf{x}_{t_{\text{out}}})\|$$

$$\lesssim \frac{\ell\kappa^3}{\lambda} + \|\mathcal{G}_{\mathcal{L}_\lambda^*,\eta}(\mathbf{x}_{t_{\text{out}}})\|$$

$$\leq \frac{\alpha}{2} + \|\mathcal{G}_{\mathcal{L}_\lambda^*,\eta}(\mathbf{x}_{t_{\text{out}}})\| , \tag{9}$$

where the second inequality is due to Lemma 2.4, the third due to Lemma 3.3.b, and the last by our assignment of $\lambda$. It therefore remains to bound $\|\mathcal{G}_{\mathcal{L}_\lambda^*,\eta}(\mathbf{x}_{t_{\text{out}}})\|$.

To that end, applying Proposition 3.6 to the function $h = \mathcal{L}_\lambda^*$, under our assignment of $T$ — which accounts for the smoothness and initial sub-optimality bounds ensured by Lemma 3.3 — we get that $\|\mathcal{G}_{\mathcal{L}_\lambda^*,\eta}(\mathbf{x}_{t_{\text{out}}})\| \leq \frac{\alpha}{2}$, for $\alpha$ as small as

$$\alpha = \Theta\left(\max\{\beta, \sigma\sqrt{d_x \log(T/\gamma)}\}\right) \tag{10}$$

By Lemma 3.4, it holds that

$$\beta = \widetilde{\mathcal{O}}\left(\frac{\lambda \ell \kappa \sqrt{d_y T}}{\epsilon n}\right) = \widetilde{\mathcal{O}}\left(\frac{\ell^{3/2}\kappa^{11/2}\Delta_F^{1/2}\sqrt{d_y}}{\alpha^2 \epsilon n}\right) \tag{11}$$

and we also have

$$\sigma\sqrt{d_x \log(T/\gamma)} = \widetilde{\mathcal{O}}\left(\frac{\ell^{3/2}\kappa^{5/2}\Delta_F^{1/2}\sqrt{d_x}}{\alpha \epsilon n}\right) . \tag{12}$$

Plugging (11) and (12) back into Eq. (10), and solving for $\alpha$, completes the proof.

## A.5   PROOF OF THEOREM 4.1

Throughout this section, we abbreviate $b = b_{\text{out}}$. We will need the following lemma, which is the mini-batch analogue of Lemma 3.4 from the full-batch setting.

**Lemma A.8.** *If* $\lambda \geq \max\{\frac{2L_1^g}{\mu_g}, \frac{L_0^f}{L_0^g}, \frac{L_1^f}{L_1^g}\}$, *then there is* $\beta_b = \widetilde{\mathcal{O}}\left(\frac{\lambda \ell \kappa \sqrt{d_y T}}{\epsilon n} + \frac{\ell \kappa}{b}\right)$ *such that with probability at least* $1 - \gamma/2$, $\mathbf{g}_t^B := \nabla_x f(\mathbf{x}_t, \widetilde{\mathbf{y}}_t^\lambda; B_t) + \lambda\left(\nabla_x g(\mathbf{x}_t, \widetilde{\mathbf{y}}_t^\lambda; B_t) - \nabla_x g(\mathbf{x}_t, \widetilde{\mathbf{y}}_t; B_t)\right)$ *satisfies for all* $t \in \{0, \ldots, T-1\} :$ $\|\nabla\mathcal{L}_\lambda^*(\mathbf{x}_t) - \mathbf{g}_t^B\| \leq \beta_b$.

*Proof of Lemma A.8.* It holds that

$$\left\|\nabla\mathcal{L}_\lambda^*(\mathbf{x}_t) - \mathbf{g}_t^B\right\| \leq \left\|\nabla\mathcal{L}_\lambda^*(\mathbf{x}_t) - \mathbb{E}[\mathbf{g}_t^B]\right\| + \left\|\mathbf{g}_t^B - \mathbb{E}[\mathbf{g}_t^B]\right\| .$$

To bound the first summand, note that $\mathbb{E}[\mathbf{g}_t^B] = \nabla_x f(\mathbf{x}_t, \widetilde{\mathbf{y}}_t^\lambda) + \lambda\left(\nabla_x g(\mathbf{x}_t, \widetilde{\mathbf{y}}_t^\lambda) - \nabla_x g(\mathbf{x}_t, \widetilde{\mathbf{y}}_t)\right)$, and therefore with probability at least $1 - \gamma/4 :$

$$\left\|\nabla\mathcal{L}_\lambda^*(\mathbf{x}_t) - \mathbb{E}[\mathbf{g}_t^B]\right\| = \mathcal{O}\left(\frac{\lambda L_1^g L_0^g \sqrt{d_y T}}{\epsilon \mu_g n}\right) = \mathcal{O}\left(\frac{\lambda \ell \kappa \sqrt{d_y T}}{\epsilon n}\right) ,$$

following the same proof as Lemma 3.4 in Appendix A.1, by replacing Theorem A.2 by the mini-batch Theorem B.1 (whose guarantee holds regardless of the inner batch size) .

To bound the second summand, note that $\|\nabla_x f(\mathbf{x}_t, \widetilde{\mathbf{y}}_t^\lambda; \xi) + \lambda(\nabla_x g(\mathbf{x}_t, \widetilde{\mathbf{y}}_t^\lambda; \xi) - \nabla_x g(\mathbf{x}_t, \widetilde{\mathbf{y}}_t; \xi))\| \leq M = \mathcal{O}(\ell\kappa)$ for every $\xi \in \Xi$, by Lemma 3.5. Hence, $\mathbf{g}_t^B$ is the average of $b$ independent vectors bounded by $M$, all with the same mean, and therefore a standard concentration bound (cf. Jin et al. 2019) ensures that $\|\mathbf{g}_t^B - \mathbb{E}[\mathbf{g}_t^B]\| = \widetilde{\mathcal{O}}(M/b)$ with probability at least $1 - \gamma/4$, which completes the proof. $\qquad\square$

We can now prove the main mini-batch result:

*Proof of Theorem 4.1.* We start by proving the privacy guarantee. Since $\mathcal{L}_{\lambda,i}^*$ is $\mathcal{O}(\ell\kappa)$-Lipschitz by Lemma 3.5, the sensitivity of $\nabla_x f(\mathbf{x}_t, \widetilde{\mathbf{y}}_t^\lambda; B_t) + \lambda\left(\nabla_x g(\mathbf{x}_t, \widetilde{\mathbf{y}}_t^\lambda; B_t) - \nabla_x g(\mathbf{x}_t, \widetilde{\mathbf{y}}_t; B_t)\right)$ is at most $\mathcal{O}(\ell\kappa)$. Accordingly, the "unamplified" Gaussian mechanism (Lemma 2.2) ensures $(\tilde{\epsilon}, \tilde{\delta})$-DP for $\tilde{\epsilon} = \widetilde{\Theta}\left(\frac{\ell\kappa}{b\sigma}\right)$, and hence is amplified (Lemma 2.3) to $(\epsilon_0, \delta_0)$-DP for $\epsilon_0 = \widetilde{\Theta}\left(\frac{\ell\kappa}{b\sigma} \cdot \frac{b}{n}\right) = \frac{\epsilon}{\sqrt{18T}}$, the last holding for sufficiently large $\sigma^2 = \widetilde{\Theta}\left(\frac{\ell^2\kappa^2 T}{\epsilon^2 n^2}\right)$, and for $\delta_0 = \frac{\delta}{3(T+1)}$. Therefore, basic composition shows that each iteration of the algorithm is $(3 \cdot \frac{\epsilon}{\sqrt{18T}}, 3\frac{\delta}{3(T+1)}) = (\frac{\epsilon}{\sqrt{2T}}, \frac{\delta}{T+1})$-DP.

Since $\epsilon/\sqrt{T} \ll 1$ under our parameter assignment, advanced composition over the $T$ iterations yields the $(\epsilon, \delta)$-DP guarantee.

We turn to analyze the utility of the algorithm. It holds that

$$
\begin{aligned}
\left\| \mathcal{G}_{F,\eta}(\mathbf{x}_{t_{\text{out}}}) \right\| &\leq \left\| \mathcal{G}_{F,\eta}(\mathbf{x}_{t_{\text{out}}}) - \mathcal{G}_{\mathcal{L}_\lambda^*,\eta}(\mathbf{x}) \right\| + \left\| \mathcal{G}_{\mathcal{L}_\lambda^*,\eta}(\mathbf{x}_{t_{\text{out}}}) \right\| \\
&\leq \left\| \nabla F(\mathbf{x}_{t_{\text{out}}}) - \nabla \mathcal{L}_\lambda^*(\mathbf{x}) \right\| + \left\| \mathcal{G}_{\mathcal{L}_\lambda^*,\eta}(\mathbf{x}_{t_{\text{out}}}) \right\| \\
&\lesssim \frac{\ell \kappa^3}{\lambda} + \left\| \mathcal{G}_{\mathcal{L}_\lambda^*,\eta}(\mathbf{x}_{t_{\text{out}}}) \right\| \\
&\leq \frac{\alpha}{2} + \left\| \mathcal{G}_{\mathcal{L}_\lambda^*,\eta}(\mathbf{x}_{t_{\text{out}}}) \right\| ,
\end{aligned}
\tag{13}
$$

where the second inequality is due to Lemma 2.4, the third due to Lemma 3.3, and the last by our assignment of $\lambda$. It therefore remains to bound $\left\| \mathcal{G}_{\mathcal{L}_\lambda^*,\eta}(\mathbf{x}_{t_{\text{out}}}) \right\|$.

To that end, applying Proposition 3.6 to the function $h = \mathcal{L}_\lambda^*$, under our assignment of $T$ — which accounts for the smoothness and initial sub-optimality bounds ensured by Lemma 3.3 — we get that $\left\| \mathcal{G}_{\mathcal{L}_\lambda^*,\eta}(\mathbf{x}_{t_{\text{out}}}) \right\| \leq \frac{\alpha}{2}$, for $\alpha$ as small as

$$
\alpha = \Theta \left( \max\{\beta_b, \sigma\sqrt{d_x \log(T/\gamma)}\} \right)
\tag{14}
$$

By Lemma A.8, it holds that

$$
\beta_b = \widetilde{\mathcal{O}} \left( \frac{\lambda \ell \kappa \sqrt{d_y T}}{\epsilon n} + \frac{\ell\kappa}{b} \right) = \widetilde{\mathcal{O}} \left( \frac{\ell^{3/2} \kappa^{11/2} \Delta_F^{1/2} \sqrt{d_y}}{\alpha^2 \epsilon n} + \frac{\ell\kappa}{b} \right) ,
\tag{15}
$$

and we also have

$$
\sigma\sqrt{d_x \log(T/\gamma)} = \widetilde{\mathcal{O}} \left( \frac{\ell^{3/2} \kappa^{5/2} \Delta_F^{1/2} \sqrt{d_x}}{\alpha \epsilon n} \right)
\tag{16}
$$

Plugging (15) and (16) back into Eq. (14), and solving for $\alpha$, completes the proof.

$\square$

## A.6 PROOF OF THEOREM 5.1

We first need a simple lemma that immediately follows from our assumptions, and Eq. (1):

**Lemma A.9.** *Under Assumptions 2.5 and 2.6, $F(\cdot\,;\xi)$ is $G$-Lipschitz, for $G = \mathcal{O}(\ell\kappa)$.*

Accordingly, the main tool that will allow us to obtain generalization guarantees, is the following uniform convergence result in terms of gradients:

**Lemma A.10** (Mei et al., 2018, Theorem 1). *Suppose $\overline{\mathcal{X}} \subset \mathbb{R}_x^d$ is a subset of bounded diameter $\text{diam}(\overline{\mathcal{X}}) \leq D$, and that $S \sim \mathcal{P}^n$. Then with probability at least $1 - \gamma$ for all $\mathbf{x} \in \overline{\mathcal{X}}$: $\|\nabla F_{\mathcal{P}}(\mathbf{x}) - \nabla F_S(\mathbf{x})\| = \widetilde{\mathcal{O}} \left( G\sqrt{d_x \log(D/\gamma)/n} \right)$, where $G$ is the Lipschitz constant of $F$.*[5]

*Proof of Theorem 5.1.* With probability at least $1 - \gamma/2$ it holds that

$$
\begin{aligned}
\left\| \mathcal{G}_{F_{\mathcal{P}},\eta}(\mathbf{x}_{\text{out}}) \right\| &\leq \left\| \mathcal{G}_{F_S,\eta}(\mathbf{x}_{\text{out}}) \right\| + \left\| \mathcal{G}_{F_{\mathcal{P}},\eta}(\mathbf{x}_{\text{out}}) - \mathcal{G}_{F_S,\eta}(\mathbf{x}_{\text{out}}) \right\| \\
&\leq \left\| \mathcal{G}_{F_S,\eta}(\mathbf{x}_{\text{out}}) \right\| + \left\| \nabla F_{\mathcal{P}}(\mathbf{x}_{\text{out}}) - \nabla F_S(\mathbf{x}_{\text{out}}) \right\| \\
&= \left\| \mathcal{G}_{F_S,\eta}(\mathbf{x}_{\text{out}}) \right\| + \widetilde{\mathcal{O}} \left( \ell\kappa\sqrt{\frac{d_x}{n}} \right) ,
\end{aligned}
$$

where the second inequality is due to Lemma 2.4, and the last is by Lemma A.10 with the Lipschitz bound of Lemma A.9 and the domain bound $\|\mathbf{x}_{\text{out}} - \mathbf{x}_0\| \leq D$ for some sufficiently large $D$ which is polynomial in all problem parameters (therefore only affecting log terms). The results then follow from Theorems 3.1 and 4.1. $\square$

---

[5] Mei et al. (2018) originally stated this for functions whose gradients are sub-Gaussian vectors. By Lemma A.9, the gradients are $G$-bounded, hence $\mathcal{O}(G)$-sub-Gaussian.

## B  OPTIMAL DP ALGORITHM FOR STRONGLY-CONVEX OBJECTIVES

The goal of this appendix is to provide a self contained analysis of a DP algorithm for strongly-convex optimization which achieves the optimal convergence rate with a a high probability guarantee. Any such algorithm can be used as the inner loop in our DP bilevel algorithm.

In particular, we analyze *localized* DP (S)GD. Although it would have been more natural to apply DP-SGD, this seems (at least according to our analysis) to yield an inferior rate with respect to the required high probability guarantee.[6] Indeed, for DP-(S)GD, previous works (such as Bassily et al. 2014) typically provide bounds in expectation, and then convert them into high-probability bounds via a black-box reduction, which applies several runs and selects the best run via the private noisy-min (i.e. Laplace mechanism). The additional error incurred by this selection is of order $\frac{1}{n}$, which translates to $\frac{1}{\sqrt{n}}$ in terms of distance to the optimum, thus spoiling the fast rate of $\frac{1}{n}$ otherwise achieved in expectation for strongly-convex objectives. We therefore resort to localization (Feldman et al., 2020): by running projected-(S)GD over balls with shrinking radii, applying martingale concentration bounds enables us to show that the distance to optimum shrinks as $R_{m+1} \lesssim \sqrt{\frac{R_m}{n}} + \frac{1}{n}$, and thus with negligible overhead we eventually recover the optimal fast rate $R_M \lesssim \frac{1}{n}$ with high probability. Our analysis differs than previous localization analyses, as it does not require adapting the noise-level and step sizes throughout the rounds.

We prove the following (which easily implies also the full-batch Theorem A.2):

**Theorem B.1.** *Suppose that $h : \mathbb{R}^{d_y} \times \Xi \to \mathbb{R}$ is a $\mu$-strongly-convex function of the form $h(\mathbf{y}) = \frac{1}{n} \sum_{i=1}^{n} h(\mathbf{y}, \xi_i)$ where $h(\cdot, \xi_i)$ is $L$-Lipschitz for all $i \in [n]$. Suppose $\arg \min h =: \mathbf{y}^* \in \mathbb{B}(\mathbf{y}_0, R_0)$, and that $n \geq \frac{L R_0^{\frac{2}{\log(d_y)}}}{\mu \epsilon'}$. Then given any batch size $b \in \{1, \ldots, n\}$, there is an assignment of parameters $M = \log_2 \log(\frac{\mu \epsilon' n}{L})$, $\sigma_{\mathrm{SGD}}^2 = \widetilde{\mathcal{O}}\left(\frac{L^2}{\epsilon'^2}\right)$, $\eta_t = \frac{1}{\mu(t+1)}$, $T_{\mathrm{SGD}} = n^2$, $R_m = \widetilde{\Theta}\left(\sqrt{\frac{R_{m-1} L}{\mu \epsilon' n}} + \frac{L \sqrt{d_y}}{\mu \epsilon' n}\right)$ such that running Algorithm 4 satisfies $(\epsilon, \delta)$-DP, and outputs $\mathbf{y}_{\mathrm{out}}$ such that $\|\mathbf{y}_{\mathrm{out}} - \mathbf{y}^*\| = \widetilde{\mathcal{O}}\left(\frac{L \sqrt{d_y}}{\mu n \epsilon}\right)$ with probability at least $1 - \gamma$.*

*Proof of Theorem B.1.* We start by proving the privacy guarantee. By the Lipschitz assumption, the sensitivity of $\nabla h(\cdot; B_t)$ is at most $\frac{2L}{b}$, thus the "unamplified" Gaussian mechanism (Lemma 2.2) ensures $(\tilde{\epsilon}, \tilde{\delta})$-DP with $\tilde{\epsilon} = \widetilde{\Theta}\left(\frac{L}{b\sigma}\right) = \widetilde{\Theta}\left(\frac{\epsilon'}{b}\right)$, and hence is amplified (Lemma 2.3) to $(\epsilon_0, \delta_0)$-DP for $\epsilon_0 = \widetilde{\Theta}\left(\frac{\epsilon'}{b} \cdot \frac{b}{n}\right) = \widetilde{\Theta}\left(\frac{\epsilon'}{n}\right) = \widetilde{\Theta}\left(\frac{\epsilon'}{\sqrt{T}}\right)$. Advanced composition (Lemma 2.1) therefore ensures that the overall algorithm is $(\epsilon', \delta')$-DP (note that this uses the fact that $M = \widetilde{\mathcal{O}}(1)$).

We turn to prove the utility of the algorithm. We first show that for all $m$ :

$$\Pr\left[\mathbf{y}^* \in \mathbb{B}(\mathbf{y}_0^m, R_m)\right] \geq 1 - \frac{m\gamma}{M} . \tag{17}$$

We prove this by induction over $m$. The base case $m = 0$ follows by the assumption $\mathbf{y}^* \in \mathbb{B}(\mathbf{y}_0, R_0)$. Denoting $\mathbf{e}_t^m := \nabla h(\mathbf{y}_t^m; B_t) - \nabla h(\mathbf{y}_t^m)$, using the inductive hypothesis that $\mathbf{y}^* \in \mathbb{B}(\mathbf{y}_0^m, R_m)$ with probability at least $1 - \frac{m\gamma}{M}$, under this probably event we get

---

[6]Even if we would have sought only expectation bounds with respect to the hyperobjective, the high probability bound with respect to the inner problem is key to being able to argue about the gradient inexactness thereafter.

$$\left\| \mathbf{y}_{t+1}^m - \mathbf{y}^* \right\|^2 = \left\| \text{Proj}_{\mathbb{B}(\mathbf{y}_0^m, R_m)} \left[ \mathbf{y}_t^m - \eta_t (\nabla h(\mathbf{y}_t^m; B_t^m) + \nu_t^m) \right] - \mathbf{y}^* \right\|^2$$

$$\leq \left\| \mathbf{y}_t^m - \eta_t (\nabla h(\mathbf{y}_t^m; B_t^m) + \nu_t^m) - \mathbf{y}^* \right\|^2$$

$$= \left\| \mathbf{y}_t^m - \mathbf{y}^* \right\|^2 - 2\eta_t \langle \mathbf{y}_t^m - \mathbf{y}^*, \nabla h(\mathbf{y}_t^m; B_t^m) + \nu_t^m \rangle + \eta_t^2 \left\| \nabla h(\mathbf{y}_t^m; B_t^m) + \nu_t^m \right\|^2$$

$$\leq \left\| \mathbf{y}_t^m - \mathbf{y}^* \right\|^2 - 2\eta_t \langle \mathbf{y}_t^m - \mathbf{y}^*, \nabla h(\mathbf{y}_t) + \mathbf{e}_t^m + \nu_t^m \rangle + 2\eta_t^2 \left( \left\| \nu_t^m \right\|^2 + \left\| \nabla h(\mathbf{y}_t) \right\|^2 \right)$$

$$= \left\| \mathbf{y}_t^m - \mathbf{y}^* \right\|^2 - 2\eta_t \langle \mathbf{y}_t^m - \mathbf{y}^*, \nabla h(\mathbf{y}_t^m) \rangle$$

$$- 2\eta_t \langle \mathbf{y}_t^m - \mathbf{y}^*, \mathbf{e}_t^m + \nu_t^m \rangle + 2\eta_t^2 \left( \left\| \nu_t^m \right\|^2 + \left\| \nabla h(\mathbf{y}_t) \right\|^2 \right) .$$

Rearranging, and using the strong convexity and Lipschitz assumptions, we see that

$$h(\mathbf{y}_t^m) - h(\mathbf{y}^*) \leq \langle \mathbf{y}_t^m - \mathbf{y}^*, \nabla h(\mathbf{y}_t^m) \rangle - \frac{\mu}{2} \left\| \mathbf{y}_t^m - \mathbf{y}^* \right\|^2$$

$$\leq \left( \frac{1}{2\eta_t} - \frac{\mu}{2} \right) \left\| \mathbf{y}_t^m - \mathbf{y}^* \right\|^2 - \frac{1}{2\eta_t} \left\| \mathbf{y}_{t+1}^m - \mathbf{y}^* \right\|^2$$

$$- \langle \mathbf{y}_t^m - \mathbf{y}^*, \mathbf{e}_t^m + \nu_t^m \rangle + \eta_t \left( \left\| \nu_t^m \right\|^2 + L^2 \right) .$$

Averaging over $t$ and using $\eta_t = \frac{1}{\mu(t+1)}$, which satisfies $\left( \frac{1}{\eta_t} - \frac{1}{\eta_{t-1}} - \mu \right) \leq 0$ and $\frac{1}{T} \sum_{t=0}^T \eta_t \lesssim \frac{\log T}{\mu T}$, by Jensen's inequality, overall we get with probability at least $1 - \frac{m\gamma}{M}$ :

$$h(\mathbf{y}_0^{m+1}) - h(\mathbf{y}^*) = h \left( \frac{1}{T} \sum_{t=0}^{T-1} \mathbf{y}_t^m \right) - h(\mathbf{y}^*)$$

$$\lesssim \underbrace{\left| \frac{1}{T} \sum_{t=0}^{T-1} \langle \mathbf{y}_t^m - \mathbf{y}^*, \mathbf{e}_t^m + \nu_t^m \rangle \right|}_{(I)} + \frac{L^2 \log T}{\mu T} + \frac{\log T}{\mu T} \underbrace{\sum_{t=0}^{T-1} \left\| \nu_t^m \right\|^2}_{(II)} . \quad (18)$$

We now apply concentration inequalities to bound $(I)$ and $(II)$ with high probability, for which we will use basic properties of sub-Gaussian distributions (cf. Vershynin 2018, §3.4). To bound $(I)$, note that for all $t$ : $\mathbb{E}\mathbf{e}_t^m = \mathbb{E}\nu_t^m = \mathbf{0}$ and therefore $\mathbb{E} \langle \mathbf{y}_t^m - \mathbf{y}^*, \mathbf{e}_t^m + \nu_t^m \rangle = 0$. Moreover, $\mathbf{e}_t^m = \frac{1}{b} \sum_{\xi \in B_t^m} (\nabla h(\mathbf{x}_t^m; \xi) - \nabla h(\mathbf{y}_t^m))$ is the average of $b$ independent vectors with norm bounded by at most $2L$, while $\nu_t^m \sim \mathcal{N}(\mathbf{0}, \sigma_{\text{SGD}}^2 I_{d_y}) = \mathcal{N}(\mathbf{0}, \widetilde{\mathcal{O}}(\frac{L^2}{\epsilon'^2}) \cdot I_{d_y})$, and also $\left\| \mathbf{y}_t^m - \mathbf{y}^* \right\| \leq R_m$ by the inductive hypothesis. By combining all of these observations, we see that $\langle \mathbf{y}_t^m - \mathbf{y}^*, \mathbf{e}_t^m + \nu_t^m \rangle$ is a $\mathcal{O}(R_m \cdot (\frac{L}{b} + \frac{L}{\epsilon'})) = \mathcal{O}(\frac{R_m L}{\epsilon'})$-sub-Gaussian random variable. By Azuma's inequality for sub-Gaussians (Shamir, 2011), we get that with probability at least $1 - \frac{\gamma}{2M}$ :

$$(I) = \widetilde{\mathcal{O}} \left( \frac{\frac{R_m L}{\epsilon'} \log(\gamma/M)}{\sqrt{T}} \right) = \widetilde{\mathcal{O}} \left( \frac{R_m L}{\epsilon' n} \right) . \quad (19)$$

To bound $(II)$, by concentration of the Gaussian norm (cf. Vershynin 2018, Theorem 3.1.1) and the union bound we can get that with probability at least $1 - \frac{\gamma}{2M}$ :

$$(II) = \widetilde{\mathcal{O}} \left( d_y \sigma_{\text{SGD}}^2 \right) = \widetilde{\mathcal{O}} \left( \frac{d_y L^2}{\epsilon'^2} \right) . \quad (20)$$

Plugging Eqs. (19) and (20) into Eq. (18), we overall get that with probability at least $1 - \frac{m\gamma}{M} - 2 \cdot \frac{\gamma}{2M} = 1 - \frac{(m+1)\gamma}{M}$ :

$$h(\mathbf{y}_0^{m+1}) - h(\mathbf{y}^*) = \widetilde{\mathcal{O}} \left( \frac{R_m L}{\epsilon' n} + \frac{d_y L^2}{\mu \epsilon'^2 n^2} \right) .$$

Applying the $\mu$-strong-convexity of $h$, and sub-additivity of the square root, we get that

$$\left\| \mathbf{y}_0^{m+1} - \mathbf{y}^* \right\| \leq \sqrt{\frac{2(h(\mathbf{y}_0^{m+1}) - h(\mathbf{y}^*))}{\mu}} = \widetilde{\mathcal{O}} \left( \sqrt{\frac{R_m L}{\mu \epsilon' n}} + \frac{L \sqrt{d_y}}{\mu \epsilon' n} \right) \leq R_{m+1} .$$

We have therefore proven Eq. (17). In particular for $m = M$ we get that with probability at least $1 - \gamma$ :

$$\|\mathbf{y}_{\text{out}} - \mathbf{y}^*\| \leq R_M \, , \tag{21}$$

hence it remains to bound $R_M$. We will prove, once again by induction over $m$, that

$$R_m = \widetilde{\mathcal{O}} \left( R_0^{\frac{1}{2^m}} \left( \frac{L}{\mu\epsilon'n} \right)^{1 - \frac{1}{2^m}} + \frac{L}{\mu\epsilon'n} \sum_{i=1}^{m} d_{\hat{y}}^{\frac{1}{2^i}} \right) \, . \tag{22}$$

The base $m = 0$ simply follows since the left hand side in Eq. (22) reduces to $R_0$. Denoting $A := \frac{L}{\mu\epsilon'n}$, by our assignment of $R_{m+1}$, the induction hypothesis and sub-additivity of the square root we get:

$$
\begin{aligned}
R_{m+1} &= \widetilde{\mathcal{O}} \left( \sqrt{R_m A} + A\sqrt{d_y} \right) \\
&= \widetilde{\mathcal{O}} \left( A^{1/2} \left( R_0^{\frac{1}{2^{m+1}}} A^{\frac{1}{2} - \frac{1}{2^{m+1}}} + A^{1/2} \sum_{i=1}^{m} d_{\hat{y}}^{\frac{1}{2^{i+1}}} \right) + A d_y^{1/2} \right) \\
&= \widetilde{\mathcal{O}} \left( R_0^{\frac{1}{2^{m+1}}} A^{1 - \frac{1}{2^{m+1}}} + A \sum_{i=1}^{m+1} d_{\hat{y}}^{\frac{1}{2^i}} \right) \, ,
\end{aligned}
$$

therefore proving Eq. (22). In particular, for $m = M = \log_2 \log(\frac{\mu\epsilon'n}{L})$, which satisfies $\frac{1}{2^M} = \frac{1}{\log(\frac{\mu\epsilon'n}{L})} = \frac{1}{\log(1/A)}$ we get

$$
\begin{aligned}
R_M &= \widetilde{\mathcal{O}} \left( R_0^{\frac{1}{\log(1/A)}} A^{1 + \frac{1}{\log(A)}} + M A d_y^{1/2} \right) \\
&= \widetilde{\mathcal{O}} \left( R_0^{\frac{1}{\log(1/A)}} A + A d^{1/2} \right) \\
&= \widetilde{\mathcal{O}} \left( R_0^{\frac{1}{\log(\mu\epsilon'n/L)}} \frac{L}{\mu\epsilon'n} + \frac{L d_y^{1/2}}{\mu\epsilon'n} \right) \\
&= \widetilde{\mathcal{O}} \left( \frac{L\sqrt{d_y}}{\mu\epsilon'n} \right) \, ,
\end{aligned}
$$

where the last follows from our assumption on $n$. This completes the proof by Eq. (21). $\qquad\square$

## C  AUXILIARY LEMMA

**Lemma 2.4.** *For any* $\mathbf{x}, \mathbf{v}, \mathbf{w} \in \mathbb{R}^d$, $\eta > 0 :$ $\|\mathcal{G}_{\mathbf{v},\eta}(\mathbf{x}) - \mathcal{G}_{\mathbf{w},\eta}(\mathbf{x})\| \leq \|\mathbf{v} - \mathbf{u}\|$.

*Proof of Lemma 2.4.* The proof is due to Ghadimi et al. (2016). By definition,

$$\mathcal{P}_{\mathbf{v},\eta}(\mathbf{x}) = \arg\min_{\mathbf{u} \in \mathcal{X}} \left\{ \langle \mathbf{v}, \mathbf{u} \rangle + \frac{1}{2\eta} \|\mathbf{x} - \mathbf{u}\|^2 \right\} \, ,$$

$$\mathcal{P}_{\mathbf{w},\eta}(\mathbf{x}) = \arg\min_{\mathbf{u} \in \mathcal{X}} \left\{ \langle \mathbf{w}, \mathbf{u} \rangle + \frac{1}{2\eta} \|\mathbf{x} - \mathbf{u}\|^2 \right\} \, ,$$

hence by first order optimality criteria, for any $\mathbf{u} \in \mathcal{X}$ :

$$\left\langle \mathbf{v} + \frac{1}{\eta}(\mathcal{P}_{\mathbf{v},\eta}(\mathbf{x}) - \mathbf{x}), \mathbf{u} - \mathcal{P}_{\mathbf{v},\eta}(\mathbf{x}) \right\rangle \geq 0 \, ,$$

$$\left\langle \mathbf{w} + \frac{1}{\eta}(\mathcal{P}_{\mathbf{w},\eta}(\mathbf{x}) - \mathbf{x}), \mathbf{u} - \mathcal{P}_{\mathbf{w},\eta}(\mathbf{x}) \right\rangle \geq 0 \, .$$

Plugging $\mathcal{P}_{\mathbf{w},\eta}(\mathbf{x})$ into the first inequality above, and $\mathcal{P}_{\mathbf{v},\eta}(\mathbf{x})$ into the second, shows that

$$0 \leq \left\langle \mathbf{v} + \frac{1}{\eta}(\mathcal{P}_{\mathbf{v},\eta}(\mathbf{x}) - \mathbf{x}), \mathcal{P}_{\mathbf{w},\eta}(\mathbf{x}) - \mathcal{P}_{\mathbf{v},\eta}(\mathbf{x}) \right\rangle \, ,$$

$$0 \leq \left\langle \mathbf{w} + \frac{1}{\eta}(\mathcal{P}_{\mathbf{w},\eta}(\mathbf{x}) - \mathbf{x}), \mathcal{P}_{\mathbf{v},\eta}(\mathbf{x}) - \mathcal{P}_{\mathbf{w},\eta}(\mathbf{x}) \right\rangle = \left\langle -\mathbf{w} + \frac{1}{\eta}(\mathbf{x} - \mathcal{P}_{\mathbf{w},\eta}(\mathbf{x})), \mathcal{P}_{\mathbf{w},\eta}(\mathbf{x}) - \mathcal{P}_{\mathbf{v},\eta}(\mathbf{x}) \right\rangle \, .$$

Summing the two inequalities yields

$$0 \le \left\langle \mathbf{v} - \mathbf{w} + \frac{1}{\eta}(\mathcal{P}_{\mathbf{v},\eta}(\mathbf{x}) - \mathcal{P}_{\mathbf{w},\eta}(\mathbf{x})), \mathcal{P}_{\mathbf{w},\eta}(\mathbf{x}) - \mathcal{P}_{\mathbf{v},\eta}(\mathbf{x}) \right\rangle$$

$$= \langle \mathbf{v} - \mathbf{u}, \mathcal{P}_{\mathbf{w},\eta}(\mathbf{x}) - \mathcal{P}_{\mathbf{v},\eta}(\mathbf{x}) \rangle - \frac{1}{\eta} \|\mathcal{P}_{\mathbf{w},\eta}(\mathbf{x}) - \mathcal{P}_{\mathbf{v},\eta}(\mathbf{x})\|^2$$

$$\le \|\mathcal{P}_{\mathbf{w},\eta}(\mathbf{x}) - \mathcal{P}_{\mathbf{v},\eta}(\mathbf{x})\| \left( \|\mathbf{v} - \mathbf{u}\| - \frac{1}{\eta} \|\mathcal{P}_{\mathbf{w},\eta}(\mathbf{x}) - \mathcal{P}_{\mathbf{v},\eta}(\mathbf{x})\| \right) .$$

Hence,

$$\|\mathbf{v} - \mathbf{u}\| \ge \frac{1}{\eta} \|\mathcal{P}_{\mathbf{v},\eta}(\mathbf{x}) - \mathcal{P}_{\mathbf{w},\eta}(\mathbf{x})\|$$

$$= \left\| \frac{1}{\eta} (\mathcal{P}_{\mathbf{v},\eta}(\mathbf{x}) - \mathbf{x}) - \frac{1}{\eta} (\mathcal{P}_{\mathbf{w},\eta}(\mathbf{x}) - \mathbf{x}) \right\|$$

$$= \|\mathcal{G}_{\mathbf{v},\eta}(\mathbf{x}) - \mathcal{G}_{\mathbf{w},\eta}(\mathbf{x})\| .$$

$\square$

