# OpenReview forum: "Differentially Private Bilevel Optimization"
_ICLR.cc/2025/Conference — Submitted to ICLR 2025_

### Official Review · Reviewer_iiaB · 2024-11-03

**Soundness:** 3
**Presentation:** 4
**Contribution:** 3
**Rating:** 8
**Confidence:** 3

**Summary:**

The paper proposes a novel algorithm to solve differential private bilevel optimization problems, assuming that (1) the upper-level objective is smooth and Lipschitz and (2) the lower-level function is strongly convex and locally Lipschitz around optima.
Compared to existing approaches, the proposed method is fully first-order and doesn’t need assumptions on privacy parameter $(\varepsilon, \delta)$

**Strengths:**

According to the author, the proposed method is the first fully first-order DP optimization method that solves the bilevel optimization problem. The proof seems correct to me and the paper is well-organized.

**Weaknesses:**

As the authors pointed out in the discussion section, the error rate for bilevel ERM, as well as the additive factor on the inverse batch size that appeared in minibatch bilevel ERM, could potentially be improved.

**Questions:**

The proof technique assumes that the lower-level objective $g(x, y)$ is strongly convex in $y$. Can this assumption be weakened so that only the convexity of $g(x,y)$ is required?

---

> ### Author Response · Authors · 2024-11-15
> **Comment to reviewer iiaB**
>
> We thank the reviewer for their time and effort, and are encouraged by their appreciation of our contribution and presentation.
>
> Regarding the question: The reviewer raises an important, and challenging question, even without privacy.
> Based on [1,2], though it is probably impossible to prove a general result while merely assuming lower-level (LL) convexity (even without privacy), it is likely that certain growth assumptions should suffice, such as PL. Indeed, our privacy analysis essentially only requires that (1) the penalty formulation approximates the hyperobjective; and that (2) the LL problem is solvable using some private method, both should hold true for problems with LL PL.
>
> [1] On finding small hyper-gradients in bilevel optimization: Hardness results and improved analysis, Chen et al. COLT 2024
>
> [2] On Penalty Methods for Nonconvex Bilevel Optimization and First-Order Stochastic Approximation, Kwon et al. ICLR 2024

---

> > ### Comment · Reviewer_iiaB · 2024-11-16
> > **Reply**
> >
> > Thanks for answering my question. I will keep my positive score.

---

### Official Review · Reviewer_Sqzz · 2024-11-03

**Soundness:** 4
**Presentation:** 4
**Contribution:** 3
**Rating:** 8
**Confidence:** 4

**Summary:**

This paper introduces a method for differentially private bilevel optimization. In bilevel optimization, the constraint set itself is given as another optimization problem. This submission aims to produce a value with a small gradient norm (we do not assume the "upper" objective is convex).

This problem has received a lot of attention recently because of new approaches, based on penalizing/smoothing the objective, that only require first-order information. One recent paper considered bilevel optimization in the local model and assumed access to second-order information. This submission operates in the central model and only uses gradients.

The submission provides theoretical guarantees for the minimizing the norm of the empirical gradient and for the population term. It also analyzes a minibatch variant with roughly similar guarantees.

**Strengths:**

This submission provides a clear contribution. Private bilevel optimization is certainly worthy of study. The paper is written well.

**Weaknesses:**

The submission is not very deep: once the problem is stated and we've decided to following the non-private first-order penalty methods, the analysis strikes me as essentially a process of assembling the right tools and carefully applying them and tracking the error. (I don't mean to imply that this is trivial, just that the paper would appeal to a wider audience if it had new ideas for private optimization. Maybe it does, and I wasn't able to pick them up?)

**Questions:**

We get guarantees for the gradient norm but the paper calls it "ERM." Is this a standard terminology for the non-convex problem?

---

> ### Author Response · Authors · 2024-11-15
> **Comment to reviewer Sqzz**
>
> We thank the reviewer for their time and effort, and are encouraged by their appreciation of our contribution and presentation.
>
> Regarding the question: We believe that using the term “ERM” for nonconvex problems in the context of gradient norm is indeed standard (e.g., [1]). Nonetheless, we agree with the reviewer that it might be slightly misleading, and therefore will change it to “empirical” in suitable places throughout the paper.
>
> [1] Faster Rates of Convergence to Stationary Points in Differentially Private Optimization, Arora et al., ICML 2023

---

### Official Review · Reviewer_mpce · 2024-11-03

**Soundness:** 3
**Presentation:** 3
**Contribution:** 2
**Rating:** 5
**Confidence:** 3

**Summary:**

The paper presents DP algorithms for bilevel optimization problems, where upper-level objectives are smoot and the lower-level problems are smooth and strongly convex. The proposed gradient-based DP algorithms can avoid Hessian computations.

**Strengths:**

This framework can work with different inner algorithms with only dependency on its convergence rate and DP parameters.

**Weaknesses:**

While the methods outlined in the paper appear innovative, they lack a clear comparative analysis with existing methods. Fully first-order methods have already been established in non-DP settings; however, it's not apparent whether the DP version introduces significant additional complexities.

The paper lacks empirical evaluation, which is noted as future work. This omission is unconventional and limits the ability to gauge practical effectiveness. Exploring the interaction between outer and inner algorithms through experiments could yield insightful results regarding their actual performances.

The "any desired privacy" mentioned in the contributions does not have a clear meaning because:
Adjusting a parameter to achieve a specific \epsilon,\delta value is almost always possible in all DP algorithms. The algorithm can meet any pair of \epsilon,\delta pair. However, through naive application of gaussian noise. While being correct, it does not exactly produce the desired privacy. Furthermore, meeting any privacy specification doesn't necessarily imply efficiency.

**Questions:**

The closest existing result is from Chen (2024), but due to the difference in the DP frameworks (central DP vs. local DP) it is hard to draw a direct comparation. Since both DP mechanisms are achieved by adding Gaussian noise, a question remains: When the scale of noise is identical, can the performance between these methods be effectively compared?

---

> ### Author Response · Authors · 2024-11-15
> **Comment to reviewer mpce**
>
> We thank the reviewer for their time, and appreciate their constructive efforts in helping us strengthen our work.
>
> As the reviewer pointed out, the difference in DP frameworks between our work and that of Chen and Wang indeed makes it hard to draw direct comparisons.
>
> Nonetheless, we would like to emphasize that working in the central model enables us to allow the user to trade-off privacy vs. accuracy as they wish, whereas the previous result requires certain assumptions on the noise magnitudes (Assumption 3.1 in their paper) in order for the privacy/convergence analysis to carry through. Consequently, as seen throughout the previous paper, the privacy parameter is only bounded by some (large!) constant, e.g. 66 in Appendix H therein. This is not the privacy regime in which DP methods are typically applied in practice.
>
> Moreover, we would like to emphasize that our algorithm is fully first-order and avoids Hessian computations and inversions altogether, as opposed to that of Chen and Wang, which is well-known to be a bottleneck in high dimensional applications.

---

### Official Review · Reviewer_ULdr · 2024-11-04

**Soundness:** 3
**Presentation:** 3
**Contribution:** 3
**Rating:** 6
**Confidence:** 2

**Summary:**

This paper studies bilevel optimization under the central DP model. The authors leverage recent advancements in (non-private) first-order bilevel optimization and propose algorithms that cover both ERM and population loss.
The proposed algorithm avoids computing Hessian and uses only gradients, finding approximate solutions under certain conditions. Authors also show the mini-batch variant has similar convergence properties.

**Strengths:**

- The paper studies bilevel optimization under central DP, establishing first results in the area.
- It provides a mini-batch variant and addresses both ERM and population risks.
- The paper has a well organized structure.

**Weaknesses:**

See questions below

**Questions:**

Since the paper is built on recent advancements in (non-private) first-order bilevel optimization, what are the technical challenges when moving from non-private to private case?
What are the technical difficulties of the analysis of the algorithm compared to the ones for non-private bilevel optimization?

I hope to see some empircal results if possible.

---

> ### Author Response · Authors · 2024-11-15
> **Comment to reviewer ULdr**
>
> We thank the reviewer for their time and effort, and are encouraged by their appreciation of our contribution and presentation.
>
> Regarding the question:
>
> The challenge moving to the private case is to make sure that solving the inner problem does not leak information about the outer problem. Hence, as we explain in Section 3.1, the hypergradient estimator becomes inexact, whereas in the non-private the case the inner problem can be solved up to exponentially small error, and the gradient estimation is essentially exact. Accordingly, we need to make sure that the gradient errors don’t accumulate, and our analysis of the private nonconvex outer loop takes care of this (in fact, optimally, suffering from no overhead at all).
>
> An additional technical challenge is bounding the component Lipschitz constant independently of the penalty parameter, which we do in Lemma 3.5. This Lipschitz constant affects only the private analysis (by a sensitivity argument), and is not required for the non-private case, hence was not analyzed by prior work.

---

> > ### Comment · Reviewer_ULdr · 2024-11-26
> >
> > Thank you for answering my question. Inexactness of gradients is studied for general optimization problems. Are there particular challenges for bilevel optimization that make this harder to solve? (Prior literature for handling inexact hypergradients in bilevel optimization?)

---

> > > ### Author Response · Authors · 2024-11-26
> > > **Response to reviewer ULdr**
> > >
> > > We thank the reviewer for the question and for their engagement.
> > >
> > > Indeed, gradient inexactness is well studied in optimization, under several "inexactness" models, as discussed for example in [1]. Nonetheless, we are not aware of a precise statement in the literature for smooth nonconvex optimization which satisfies all of the following properties:
> > > - Gradients are inexact -- which we need in order to privatize the lower-level problem;
> > > - Gradients are stochastic -- which we need in order to privatize the upper level problem;
> > > - Constant stepsize, which is also independent of the inexactness -- which we want for optimal convergence rates, so that privacy won't overly accumulate due to composition;
> > > - Output is a single point with small gradient with high probability, as opposed to just proving that the average gradient is small -- which is important since we want to derive high probability bounds, and can't re-check the gradient norm of a uniformly sampled iterate since returning or dismissing it based on its gradient's norm is a non-private operation.
> > >
> > > A prior analysis that we are aware of in the context of inexact hypergradients is [2], which fails to satisfy the 3rd and 4th properties above and therefore does not derive an optimal rate, nor does it work with high probability (which is hard to fix under privacy, as previously mentioned).
> > >
> > > We hope this clarifies the reviewer's question, and we would also be happy to answer any further questions if needed.
> > >
> > > [1] Bertsekas, Nonlinear Programming, 2016, Section "Gradient methods with random and nonrandom errors".
> > >
> > > [2] Giovannelli et al., Inexact bilevel stochastic gradient methods for constrained and unconstrained lower-level problems, 2023, Section 4.1

---

### Meta-Review · Area_Chair_292A · 2024-12-24

**Metareview:**

Paper studies the Differentially Private optimization of smooth nonconvex-strongly-convex Bilevel problem. Authors provide DP algorithms for solving both the ERM and population level problem in both full-batch and mini-batch settings. Only other work earlier work on DP bilevel optimization requires Hessian inversion and studies provides only stricter local DP guarantees. Theoretical results appear correct, but it is noted to be straightforward even though tedious. Unfortunately, authors do not provide any empirical evaluation to validate their results which limits the practical usefulness of the manuscript.

**Additional Comments On Reviewer Discussion:**

Even though reviewers noted that the paper does not contain any empirical results to validate the theoretical results, authors did not attempt to address the concerns. Authors appear to have addressed most other questions.

---

### Decision · Program_Chairs · 2025-01-22

Reject